# Effects of Postbiotic Administration on Canine Health: A Systematic Review and Meta-Analysis

**DOI:** 10.3390/microorganisms13071572

**Published:** 2025-07-03

**Authors:** Diego Paul Bonel-Ayuso, Javier Pineda-Pampliega, Paloma Martinez-Alesón García, Montserrat Fernandez-Muela, Jesús de la Fuente, Paloma Maria Garcia Fernandez, Beatriz Isabel Redondo

**Affiliations:** 1Department of Animal Production, Faculty of Veterinary, Complutense University of Madrid, Avenida Puerta de Hierro s/n, 28040 Madrid, Spain; jpineda@ucm.es (J.P.-P.); montsefe@ucm.es (M.F.-M.); jefuente@vet.ucm.es (J.d.l.F.); bisabelr@ucm.es (B.I.R.); 2Department of Pharmaceutical and Health Sciences, Faculty of Pharmacy, CEU San Pablo University, Plaza Montepríncipe 1D, 28668 Madrid, Spain; paloma.martinez2@ceu.es; 3Department of Animal Medicine and Surgery, Faculty of Veterinary, Complutense University of Madrid, Avenida Puerta de Hierro s/n, 28040 Madrid, Spain; garciap@vet.ucm.es

**Keywords:** microbial metabolites, microbiota, postbiotics, probiotics, antimicrobial resistance, dogs

## Abstract

Postbiotics—defined in 2021 by the International Scientific Association of Probiotics and Prebiotics (ISAPP) as preparations of inactivated microorganisms and/or their components that confer health benefits to the host—are a promising tool in veterinary medicine. This systematic review and meta-analysis evaluated their types, mechanisms of action, and physiological effects in dogs. A literature search was conducted in PubMed, Scopus, and Web of Science up to 10 October 2024. Eligible studies included peer-reviewed trials in dogs or mechanistic studies on postbiotics; studies in other species or without peer review were excluded. Risk of bias was assessed, and random-effects meta-analyses were performed when appropriate. Of 157 records, 69 met the inclusion criteria, including 13 in vivo studies in dogs. Meta-analyses of selected outcomes showed no statistically significant differences between postbiotic and control groups. Evidence is limited by small sample sizes, strain heterogeneity, and varied study designs. Despite nonsignificant results, existing evidence from other species suggests that postbiotics improve the gut microbiota composition, modulate immune and inflammatory responses, reduce oxidative stress, and aid in the treatment of chronic conditions such as atopic dermatitis. Taken together with their potential role as an alternative to antimicrobial use, these findings highlight the need for further research in canine health to support the use of postbiotics in the treatment of common canine diseases, either as a standalone therapy or in combination with existing therapeutic options.

## 1. Introduction

### 1.1. What Is the Gut Microbiota?

In recent years, interest in the relationship between an individual and the microorganisms residing in their gut has grown significantly [1]. The gut microbiota consists of a vast array of bacteria, fungi, viruses, archaea, and protozoa that inhabit the host’s gastrointestinal tract, establishing a symbiotic relationship [2]. This diverse microbial community plays a crucial role in canine health and immunity, influencing a wide range of physiological functions, including nutrient digestion, vitamin synthesis, immune response modulation, and protection against pathogens [3].

The composition of the human gut microbiota varies along the gastrointestinal tract. Microbial populations are sparse in the stomach and small intestine, whereas the large intestine harbors approximately 10^12^ bacterial cells per gram, encompassing between 300 and 1000 distinct species, most of which are anaerobic. The predominant phyla in the human gut microbiota include Firmicutes, Bacteroidetes, Actinobacteria, and Proteobacteria. The most representative bacterial genera are *Bacteroides*, *Clostridium*, *Peptococcus*, *Bifidobacterium*, *Eubacterium*, *Ruminococcus*, *Faecalibacterium*, and *Peptostreptococcus* [4]. Microbial diversity reaches its peak when Bacteroidetes account for 15% of the total population and Firmicutes comprise 80%. Additionally, the presence of Tenericutes, Euryarchaeota, Lentisphaerae, and Cyanobacteria has been associated with increased diversity, whereas Proteobacteria and Fusobacteria are linked to lower diversity levels [5]. In-depth analysis of the microbiota composition and structural differences between healthy and diseased individuals has led to the development of microbiota-targeted therapeutic strategies for various medical conditions [6].

The composition of the gut microbiota is currently one of the most dynamic fields of scientific research due to its profound impact on human health [1]. Microbiota balance is often evaluated through overall diversity, commonly assessed using alpha and beta diversity metrics. These parameters quantify differences in microbial composition between individuals or groups [5]. Alpha diversity measures microbiota richness (the number of taxonomic groups present) and evenness (the relative abundance of each taxonomic group) within a single sample or population. In contrast, beta diversity assesses compositional differences between distinct groups or samples [7].

A lower alpha diversity has been associated with the onset of both acute and chronic diseases [5], particularly inflammatory and autoimmune disorders such as rheumatoid arthritis, inflammatory bowel disease, systemic lupus erythematosus, and multiple sclerosis [8]. Certain diseases, such as colorectal cancer, are linked to an increased presence of pathogenic bacteria (*Fusobacterium*, *Porphyromonas*, *Peptostreptococcus*, *Parvimonas*, and *Enterobacter*). In contrast, conditions such as inflammatory bowel disease are more prevalent in individuals with a reduced abundance of beneficial bacterial families (Ruminococcaceae and Lachnospiraceae) within their gut microbiota [9].

Like humans, animals and the environment also harbor a highly diverse microbiota, whose composition significantly influences host health. Human, animal, and environmental health are deeply interconnected, with each relying on the others [10]. As part of this intricate relationship, genetic material is exchanged among the microorganisms that comprise the microbiota of these three components. This highlights the fundamental role of the microbiota in humans, animals, and the environment within the One Health framework [11].

Antimicrobial resistance (AMR) genes, present in both pathogenic and commensal bacteria within the microbiota, can be horizontally transferred to other bacterial species through conjugation, transduction, and transformation [10]. However, antibiotic residues and resistance genes enter the environment through various sources, including hospital wastewater, excreta from domestic and livestock animals, and pollutants generated during antibiotic production [11]. These factors promote the development and dissemination of antimicrobial resistance within environmental microbiota. Environmental bacteria can, in turn, colonize the gut microbiota of wild animals within an ecosystem [10]. Additionally, these microorganisms can travel long distances via aerosols, water, or wildlife, eventually returning to humans and domestic or livestock animals through food and drinking water [11]. This cycle underscores the intricate interconnection between human, animal, and environmental microbiota and their crucial role in the spread of antimicrobial resistance among these hosts [12].

In dogs, the gut microbiota is composed of a wide variety of microbial genera, with their specific distribution having a significant impact on host health. The predominant bacterial phyla in the canine gut microbiota include Firmicutes, Bacteroidetes, Proteobacteria, Fusobacteria, and Actinobacteria [1,13]. However, their composition varies across different intestinal segments, adapting to the conditions, metabolites, and physiological functions of each region [14]. Anaerobic bacteria predominate in the large intestine, whereas the small intestine harbors a similar proportion of aerobic and facultative anaerobic bacteria [15]. Bacteria belonging to the order Clostridiales are dominant in the duodenum and jejunum, while those classified under the orders Fusobacteriales and Bacteroidales are most abundant in the ileum and colon. Additionally, bacteria belonging to the order Lactobacillales are widely distributed throughout the entire intestinal tract, whereas Enterobacteriales are more prevalent in the small intestine than in the large intestine [14,15] (Figure 1).

**Figure 1 microorganisms-13-01572-f001:**
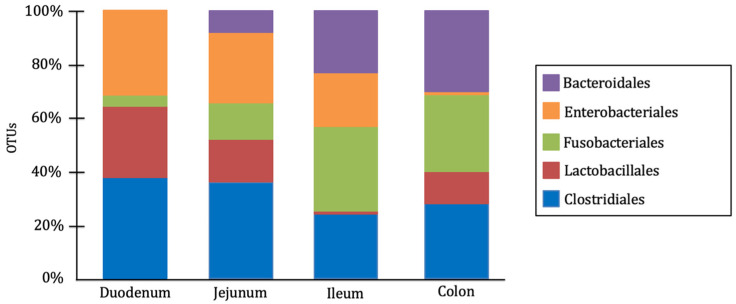
Relative abundance of major bacterial orders (Bacteroidales, Enterobacteriales, Fusobacteriales, Lactobacillales, and Clostridiales) in the microbiota of different intestinal segments (duodenum, jejunum, ileum, and colon) in healthy dogs [14].

Therefore, the canine gut microbiota consists of a diverse range of bacterial genera and other microorganisms, forming intricate interdependent relationships that contribute to a complex equilibrium essential for host health [1]. This microbial community protects the intestinal mucosa from pathogen colonization and supports gastrointestinal health through the fermentation of carbohydrates and low-digestibility proteins, participation in bile acid metabolism, and production of a wide array of metabolites involved in gastrointestinal functions and various physiological processes in other organs and systems [13].

### 1.2. Intestinal Dysbiosis and Its Effects on Canine Health

This delicate balance within the gut microbiota, closely linked to microbial diversity, can be disrupted by numerous factors, including diet, antibiotic use, host immune response, and the presence of pathogenic bacteria, among others. Such disturbances can lead to an imbalance in the microbial composition, known as dysbiosis. Intestinal dysbiosis results in impaired microbial function, reducing its contribution to nutrient digestion and the production of essential metabolites for the host [1,13]. These functional alterations in the gut microbiota can trigger acute digestive disorders, such as diarrhea, and predispose dogs to chronic digestive conditions like inflammatory bowel disease (IBD) [15].

In general, dysbiosis increases intestinal epithelial permeability, facilitating the translocation of inflammatory compounds, such as lipopolysaccharide (LPS), into the bloodstream. This, in turn, triggers the release of pro-inflammatory cytokines, including IL-1 and TNF-α. If this condition persists chronically, prolonged inflammation may lead to the development of systemic allergic and immune-mediated diseases, as well as structural damage to organs, such as joints and blood vessels [16]. Due to the close relationship between the microbiota and the rest of the body (Figure 2), alterations in microbial composition have been associated with various canine diseases, including atopic dermatitis [17], osteoarthritis [18], behavioral disorders [3,19], and chronic kidney disease [20].

**Figure 2 microorganisms-13-01572-f002:**
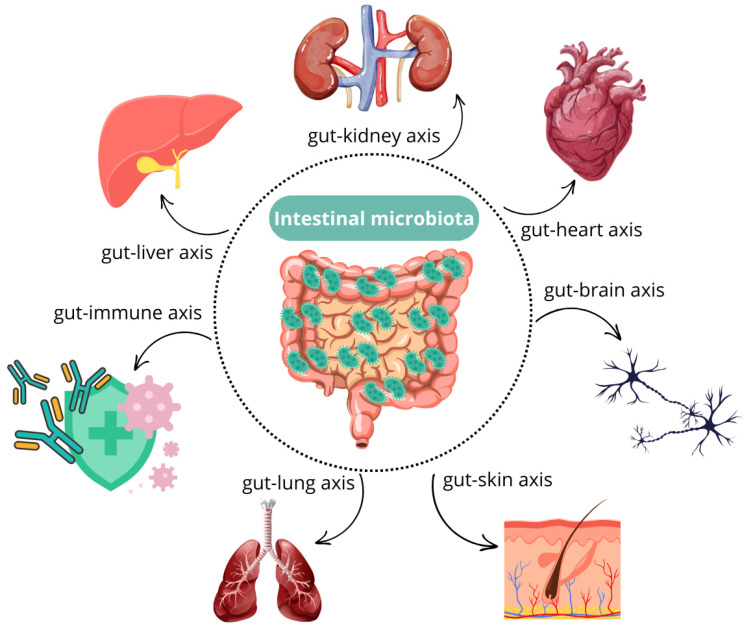
Systemic influence of the gut microbiota. The gut microbiota interacts with multiple body systems, contributing to immune regulation [16,17,21], neurochemical signaling [3,19], metabolic homeostasis [1,13], and barrier integrity [22]. These interactions affect the skin [22], immune system [17], nervous system [3,19], liver [23], cardiovascular system [24], respiratory tract [21], and urinary tract [20], underscoring the microbiota’s systemic role in health and disease.

These findings explain the observed differences in the gut microbiota composition and diversity between healthy and diseased dogs [15]. In healthy dogs, the gut microbiota is characterized by a higher abundance of bacterial species from the families Lachnospiraceae, Anaerovoracaceae, and Oscillospiraceae, as well as the genera *Ruminococcus*, *Fusobacterium*, and *Fecalibacterium* [17]. Conversely, dogs with acute diarrhea commonly exhibit a reduction in Bacteroidetes populations and in key bacterial families involved in short-chain fatty acid (SCFA) production, such as Erysipelotrichaceae (e.g., *Turicibacter*), Ruminococcaceae (e.g., *Ruminococcus* and *Faecalibacterium*), and Lachnospiraceae (e.g., *Blautia*). Additionally, these dogs show an increase in Fusobacteria, as well as the genera *Clostridium* and *Sutterella* [15].

In dogs suffering from chronic enteropathies, including IBD, a decline in *Faecalibacterium* spp. and Fusobacteria populations has been observed [15]. Dogs with atopic dermatitis present lower alpha diversity than their healthy counterparts, with increased representation of the genera *Conchiformibius*, *Catenibacterium*, and *Megamonas* [17]. Similarly, dogs with osteoarthritis exhibit a gut microbiota enriched in *Megamonas* while showing a decreased abundance of the bacterial families Paraprevotellaceae, Porphyromonadaceae, and Mogibacteriaceae [18]. Furthermore, aggressive dogs display a more diverse microbiome than healthy dogs, with a predominance of the genera *Catenibacterium* and *Megamonas* [3,19] (Figure 3).

**Figure 3 microorganisms-13-01572-f003:**
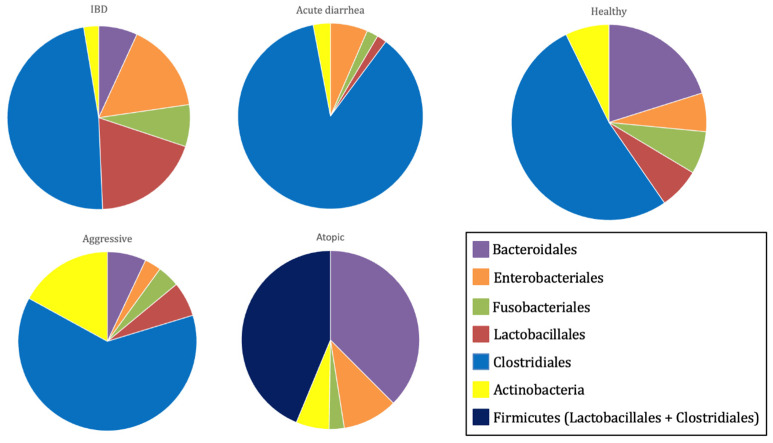
Relative abundance of major bacterial orders (Bacteroidales, Enterobacteriales, Fusobacteriales, Lactobacillales, Clostridiales, and Actinobacteria) in the fecal microbiota of healthy dogs and those with different pathologies (IBD, acute diarrhea, atopic dermatitis, and aggressive behavior) [15,17,19].

Therefore, the wide range of functions performed by the intestinal microbiota extends beyond digestive health, influencing the overall well-being of the individual (Figure 2). This has led to the identification of several physiological axes, such as the gut–brain axis, which relies on a bidirectional communication network between the enteric nervous system (ENS) and the central nervous system (CNS), with serotonin being the key neurotransmitter of this axis [3]. Due to its role in tryptophan metabolism and serotonin production, the intestinal microbiota has been shown to be essential in this axis and has been linked to the development of behavioral disorders [19].

Another important connection is the gut–lung axis, which highlights the close relationship between the microbiota in both organs. Under normal conditions, dendritic cells process antigens present in the intestine and promote T-lymphocyte proliferation in locations exposed to pathogen entry, such as the respiratory tract. However, in cases of intestinal dysbiosis, an imbalance in bacterial metabolite synthesis and an increase in pro-inflammatory mediators lead to dysregulation in the maturation and proliferation of T lymphocytes in the respiratory tract, altering immune responses in this region. As a result, intestinal dysbiosis has been associated with respiratory infections and other conditions, including asthma and allergic respiratory diseases [21].

Similarly, the gut–skin axis describes the bidirectional relationship between the microbiota of both organs. In cases of intestinal dysbiosis, increased intestinal epithelial permeability facilitates the absorption of inflammatory cytokines, whose production is also elevated. The pro-inflammatory activity of these molecules in the skin promotes the development of various conditions, such as erythematous processes, atopic dermatitis, and other allergic disorders [22].

The canine gut microbiota has also been linked to the health of other organs, notably the liver. Chronic hepatobiliary diseases are commonly associated with alterations in the bile composition, density, and flow, all of which can influence the gut microbial community. In such cases, affected dogs often exhibit an increased abundance of potentially harmful bacteria (e.g., Escherichia, Shigella, and Serratia) and a decrease in beneficial taxa such as Clostridium hiranonis and the genera Fusobacterium, Megamonas, Ruminococcus, and Turicibacter [23].

Regarding the cardiovascular system, certain metabolites produced by gut bacteria have been shown to contribute to the development of heart disease—most notably, trimethylamine N-oxide (TMAO). Dogs suffering from myxomatous mitral valve disease (MMVD) display higher levels of intestinal dysbiosis, characterized by an overrepresentation of Escherichia coli, a known TMAO-producing bacterium, thereby supporting a connection between gut microbiota and cardiac health [24].

Another example is the gut–kidney axis, in which the microbiota also plays a crucial role. Studies have shown a higher prevalence of intestinal dysbiosis in dogs with chronic kidney disease. In these patients, the bacterial composition of their intestinal microbiota leads to increased production of uremic toxins, such as indoxyl sulfate (IS) and p-cresyl sulfate (pCS), which contribute to nephron damage and the progression of kidney disease [20].

### 1.3. Modulation of Intestinal Microbiota Composition

Studies in various animal species have demonstrated that diet significantly influences the composition and diversity of the intestinal microbiota. Nevertheless, establishing a direct relationship between specific dietary components and changes in the gut microbiome remains challenging in dogs, which are considered omnivores. However, variations in nutritional components, particularly fiber quantity and type, have been associated with microbiota alterations [3,25]. Diets with higher crude protein content but lower crude fiber and nitrogen-free extract reduce the abundance of bacteria from the genera *Peptostreptococcus*, *Faecalibacterium*, *Bacteroides*, and *Prevotella*. These bacteria are involved in the fermentation of dietary fiber and carbohydrates, producing short-chain fatty acids, which play a crucial role in host health. Consequently, their reduction can have detrimental effects [3,26,27].

These findings suggest that the canine intestinal microbiota can adapt to physiological needs depending on dietary intake. However, this microbial population is also influenced by external factors, such as antibiotics [28]. Numerous adverse gastrointestinal effects have been reported following antibiotic administration, as these drugs induce intestinal dysbiosis by disrupting the host’s microbiota [28,29]. Antibiotic use significantly reduces both alpha and beta diversity, which has been linked to the development of various conditions, including immune-mediated disorders, atopic dermatitis, degenerative joint disease, and skin allergies [17,18,30,31].

The increasing recognition of the intestinal microbiota’s role has driven the field of animal nutrition towards the use of functional ingredients and bioactive compounds to modulate the gut microbial composition and enhance overall canine health (Figure 4). This approach includes prebiotics, probiotics, and synbiotics, with postbiotics recently added to the list [13,32].

Prebiotics are nutritional compounds (typically fiber based) that resist intestinal digestion, absorption, and adsorption, allowing them to be fermented by the host’s gut microbiota. This fermentation promotes the growth or activity of specific bacterial species, leading to beneficial changes in the microbiota composition, increased microbial diversity, and enhanced short-chain fatty acid production, all of which positively impact host health [33,34]. The most commonly used prebiotics in canine nutrition include lactulose, fructooligosaccharides (FOS), inulin, beta-glucans, and mannan-oligosaccharides (MOS) [35].

Probiotics are live microorganisms that have been isolated and characterized, with scientific evidence supporting their health benefits when administered in adequate amounts. Ideally, these microorganisms should be non-pathogenic to the host, capable of withstanding low pH and high bile acid concentrations (intestinal conditions), and able to adhere to the intestinal epithelium. Probiotic supplementation allows for the introduction of beneficial bacteria into the intestinal microbiota, helping to maintain microbial balance and the physiological processes it regulates [36].

Synbiotics are products that combine prebiotics and probiotics, enhancing the benefits of both while improving the survival of these supplements in the gastrointestinal tract. The most commonly used probiotics in these formulations include *Saccharomyces boulardii*, *Heyndrickxia coagulans*, and bacteria from the genera *Lactobacillus* and *Bifidobacterium*. Regarding prebiotics, fructooligosaccharides (FOS), galactooligosaccharides (GOS), xylooligosaccharides (XOS), and inulin are commonly used [37].

**Figure 4 microorganisms-13-01572-f004:**
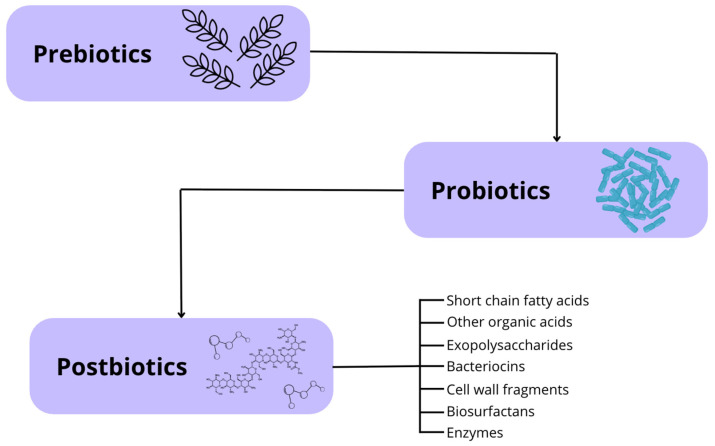
Schematic overview of prebiotics, probiotics, and postbiotics. Prebiotics are non-digestible compounds that selectively stimulate the growth or activity of beneficial gut microbes [33,34]. Probiotics are live microorganisms that confer health benefits to the host when administered in adequate amounts [36]. Postbiotics refer to bioactive compounds produced during microbial fermentation processes, including short-chain fatty acids, bacteriocins, and enzymes, which can exert beneficial effects even in the absence of live bacteria [38,39].

The term postbiotic was defined in 2021 by the International Scientific Association of Probiotics and Prebiotics (ISAPP) as a preparation of inactivated microorganisms and/or their components that provide health benefits to the host [34]. Therefore, postbiotics are soluble, bioactive metabolic products secreted by live bacteria or yeasts and released during fermentation following cell membrane lysis. Their composition includes both metabolites (enzymes, peptides, proteins, exopolysaccharides, organic acids, and lipids) and structural components of the bacterial cell wall (teichoic and lipoteichoic acids, peptidoglycans, bacterial surface layer proteins, and other polysaccharides) [38,39].

Given the limited number of studies currently available on postbiotic administration in dogs, this systematic review and meta-analysis should be regarded as an exploratory synthesis. Its primary aim is to provide an initial overview of the potential effects of postbiotics on canine health, based on the best available evidence to date. By identifying trends, knowledge gaps, and methodological challenges in this emerging field, the present study seeks to offer a foundation for more targeted and robust investigations in the future.

## 2. Materials and Methods

### 2.1. Systematic Literature Review

This review was performed in accordance with the PRISMA (Preferred Reporting Items for Systematic Reviews and Meta-Analyses) guidelines. The bibliographic search for this study was conducted using three online databases: PubMed (National Library of Medicine, NIH: https://pubmed.ncbi.nlm.nih.gov/ (accessed on 22 June 2025)), Scopus (Elsevier: https://www.scopus.com/ (accessed on 22 June 2025)), and Web of Science (Clarivate Analytics: https://www.webofscience.com/ (accessed on 22 June 2025)). The research question formulated to guide this review was: *What types of postbiotics exist, how do they exert their action, and what effects do they have on canine health?*

This systematic review and meta-analysis aimed to evaluate the effects of postbiotic administration in dogs. The review was conducted following the PICOS framework:Participants: dogs of any age and health status.Interventions: orally administered postbiotics, regardless of microbial origin.Comparisons: control or placebo groups when available.Outcomes: gut health parameters including fecal quality, microbiota composition, immune markers, and oxidative stress.Study design: randomized controlled trials or pre-post intervention studies with measurable outcomes.

To maximize the number of relevant articles for each aspect of the research question, the following keywords were used, combining different Boolean operators: [(postbiotic OR postbiotics) AND (type) AND (mechanism OR action)] OR [(dog OR dogs OR *Canis familiaris*) AND (treatment OR treatments OR therapy OR therapies OR therapeutic) AND (postbiotic OR postbiotics)].

The formal review protocol was registered on the Open Science Framework (OSF) and is available at https://doi.org/10.17605/OSF.IO/BMNAF (accessed on 30 May 2025). While the term “postbiotic” was used in the search strategy, broader terms such as “inactivated microorganisms” were excluded to maintain focus on interventions relevant to the ISAPP definition and to avoid retrieving studies from unrelated biomedical fields (e.g., vaccine trials or disinfection protocols).

To address the limitations posed by evolving terminology and database indexing, a manual cluster search strategy was also applied. This involved the targeted screening of reference lists from the initially selected studies, with the goal of identifying additional publications that matched our inclusion criteria. This strategy was especially valuable in retrieving studies describing specific postbiotic compounds and their mechanisms of action, and canine studies referenced within other dog-focused articles that were not indexed under the term postbiotic. Given the novelty of the postbiotic concept—formally defined by ISAPP only in 2021 [39]—many relevant studies used alternative terminology to describe postbiotic compounds (e.g., bacteriocins, biosurfactants, exopolysaccharides). Therefore, manual identification was essential to ensure a comprehensive and accurate selection of the eligible literature.

Using this search equation (conducted between 22 September 2024 and 10 October 2024), a total of 157 results were obtained (30 from PubMed, 59 from Scopus, and 68 from Web of Science). After removing duplicate articles, 98 unique records remained (Figure 5).

This list of articles was then reviewed according to the inclusion criteria (descriptive studies and literature reviews on postbiotic types and mechanisms of action; in vivo studies evaluating the effects of products containing at least one postbiotic component on canine health) and the exclusion criteria (in vivo studies assessing postbiotic administration in non-canine species (Reason 1); studies on postbiotics in dogs that had not undergone peer review (Reason 2); reviews and articles discussing other types of biotics (Reason 3)).

Three independent reviewers screened the titles and abstracts of all identified records to determine their eligibility according to the predefined inclusion and exclusion criteria. Full texts of potentially relevant articles were then assessed independently by the same reviewers to confirm final inclusion. For data extraction, two reviewers independently collected relevant information from each included study. Extracted data included the study design, population characteristics, postbiotic type and origin, administration protocol, outcome measures, and main results.

After applying these criteria, only 45 relevant articles remained. This reduction was because many retrieved studies focused on the mechanisms of action of prebiotics and probiotics but did not address postbiotics. Additionally, some experimental studies investigated postbiotic administration in humans but did not explore the underlying mechanisms of these metabolites.

For this reason, articles that met the inclusion criteria and were cited in the reference lists of the selected studies were also included (manual cluster searching). This approach increased the total number of relevant records to 69 (Figure 5). This methodological complement allowed for a more robust and representative overview of the available evidence.

Only 13 articles were found that evaluate the effects of postbiotic administration on canine health (Table 1). In all these studies, postbiotics were administered orally in various formats (such as tablets, powders, lozenges, or incorporated into feed). These postbiotics were obtained through the inactivation—primarily by heat—of different microbial cultures. Consequently, the final products used in these studies contained both the inactivated microorganisms and the metabolites derived from their fermentation.

**Table 1 microorganisms-13-01572-t001:** Summary of studies investigating the effects of postbiotic administration on canine health. Information has been summarized regarding the postbiotic-producing microbial species, the type of postbiotic used, the dosage, the duration of supplementation, and the main findings of each study.

Postbiotic-Producing Microorganism	Type of Postbiotics	Daily Dosage	Time of Supplementation	Effect	References
*Limosilactobacillus reuteri* NBF 1	Microbiotal cane^®^ (NBF Lanes, Milano, Italy) Tindalized bacteria and their fermentation products.	-	-	Modulate the composition of the intestinal microbiota.	[40]
		200 mg of tindalized bacterial bodies	30 days	Modulate and reduce changes in the intestinal microbiota of dogs under stress.	[41]
*Limosilactobacillus fermentum* and *Lactobacillus delbrueckii*	Heat-killed bacterial bodies along with spent fermentation medium.	6 × 10^10^ heat-killed bacterial bodies	35 days	Modulate and reduce changes in the intestinal microbiota of dogs under stress. Antioxidant action by increasing serum superoxide dismutase levels in stressed dogs. Fewer changes in serum levels of corticoid isoenzymes of alkaline phosphatase and alanine aminotransferase in stressed dogs.	[42]
*Bifidobacterium animalis* subsp. *lactis* CECT 8145	Heat-treated bacteria and their fermentation products.	10^10^ heat-treated cells	35 days	Reduction of intestinal pH in healthy dogs. Reduction in plasma levels of pancreatic polypeptide.	[43]
		10^10^ heat-treated cells	90 days	Reduction of pH and increase in propionate concentration in the feces of healthy dogs.	[44]
*Lactobacillus acidophilus*	Inactivated bacteria.	-	21 days	Increase in fecal IgA concentration in healthy dogs.	[45]
		-	-	Reduction in the production of pro-inflammatory cytokines. Reduction of serum triglyceride, cholesterol, and uric acid levels.	[46]
*Saccharomyces cerevisiae*	Profeed ADVANCED^®^ (Tereos, Moussy-le-Vieux, France) Inactivated yeast.	-	98 days	Modulate the immune system and prevent immunosenescence in geriatric dogs.	[47]
	Inactivated yeast with fermentation metabolites.	500 mg of heat-killed cell bodies	21 days	Increase leukocyte count and modulate the immune system. Modulate and reduce changes in the intestinal microbiota of dogs under stress.	[48]
	A-Max^TM^ Xtra (Arm and Hammer Animal Nutrition, Princeton, NJ, USA) Yeast grown on a media of sucrose and cane molasses, and dried with processed grain by-products.	-	63 days	Modulate and prevent immune system changes under stress. Increase production of anti-inflammatory cytokines and decrease pro-inflammatory cytokines in stressed dogs. Modulate and reduce changes in the intestinal microbiota of dogs under stress.	[49]
	EpiCor^®^ (Cargill, Wayzata, MN, USA) Inactivated yeast with fermentation metabolites.	-	70 days	Decrease the severity of lesions produced by atopic dermatitis (PVAS10 and OA-SASI indices). Modulate and reduce changes in the intestinal microbiota of dogs with atopic dermatitis.	[50]
	TruMune^®^ (Cargill, Wayzata, MN, USA) Inactivated yeast with fermentation metabolites.	783 mg of inactivated yeast	70 days	Control immune response, transepidermal water loss, and sebum concentration in dogs with atopic dermatitis. Antioxidant action by increasing serum superoxide dismutase and catalase levels.	[51]
		250 mg of inactivated yeast	21 days	Modulate and reduce changes in the intestinal microbiota of dogs under stress.	[52]

Six of these studies used postbiotics derived from *Saccharomyces cerevisiae*, while the remaining eight administered metabolites of bacterial origin. Except for the studies [50,51], which were conducted in dogs with pruritic conditions, the others were performed on healthy dogs. Five of these studies exposed the animals to stressful or physically demanding conditions to compare changes in blood parameters between postbiotic-supplemented and non-supplemented dogs.

### 2.2. Meta-Analysis

Based on the 13 identified studies evaluating the effects of postbiotic administration in dogs, several meta-analyses were conducted on selected parameters assessed in these studies. These meta-analyses aim to (i) determine the average findings regarding the effects of postbiotics on canine health, (ii) identify parameters for which postbiotic effects remain inconclusive, and (iii) establish a foundation for future research in this field.

Due to the limited number of available studies assessing the effects of postbiotics on canine health, the studies listed in Table 1 were grouped according to the specific parameters evaluated. Although the postbiotics were derived from different microorganisms, six out of the thirteen studies employed products produced by *Saccharomyces cerevisiae*, and in all cases, the postbiotic preparation involved heat inactivation of the microbial culture. Despite the heterogeneity in the target populations, the experimental design was comparable across studies. Moreover, the sample size of each study was considered when assigning statistical weight to the meta-analyses. This approach enabled an overall assessment of the effects of postbiotic administration on canine health. Nonetheless, as more evidence becomes available, future studies should consider stratifying the analysis according to the specific microbial origin of postbiotics and the health status of the canine populations to gain deeper insight into their mechanisms of action.

The parameters of interest included oxidative stress markers (superoxide dismutase, malondialdehyde), inflammation markers (pro-inflammatory and anti-inflammatory cytokines), cell-mediated immune response (levels of different T-lymphocyte subtypes), humoral immune response (immunoglobulin concentration), fecal characteristics (fecal score, pH, and concentrations of specific molecules such as short-chain fatty acids, phenol, indole, and ammonia), and the gut microbiota composition. However, due to the high heterogeneity in experimental design among the 13 studies, only a subset of parameters—fecal characteristics and the gut microbiota composition—had sufficient data for meta-analysis. Among the thirteen studies, only two measured different types of lymphocytes and immune cells [48,51], two studies analyzed immunoglobulin levels [42,48], two measured cytokine levels [48,49], and two quantified oxidative stress markers [48,51].

For each included study, outcome data were extracted and expressed as the mean value for both the postbiotic-supplemented and control groups, along with the corresponding standard deviation (SD) or standard error of the mean (SEM), as reported. When comparing groups, the primary effect measure used was the mean difference between groups.

Table 2 presents the parameters included in the meta-analyses and the respective studies from which data were extracted. Regarding fecal characteristics, the meta-analysis compared findings from different studies on fecal score, fecal pH, and the fecal concentrations of short-chain fatty acids (acetate, propionate, and butyrate), branched-chain fatty acids (isobutyrate, isovalerate, and valerate), phenol, and indole. However, fecal ammonia concentration could not be analyzed, as only two studies reported data for this parameter [48,52].

**Table 2 microorganisms-13-01572-t002:** Parameters analyzed in the meta-analyses, along with the studies selected for each parameter and the number of dogs in which each parameter was measured, including the number of dogs from each individual study.

Meta-Analysis Parameter	Number of Dogs		Analyzed Studies
	Control Group	Postbiotic-Supplemented Group	
Fecal score	54 (24 + 12 + 18)	54 (24 + 12 + 18)	[45,48,52]
Fecal pH	66 (12 + 24 + 12 + 18)	66 (12 + 24 + 12 + 18)	[44,45,48,52]
Fecal acetate concentration			
Fecal propionate concentration			
Fecal butyrate concentration			
Fecal isobutyrate concentration			
Fecal isovalerate concentration			
Fecal valerate concentration			
Fecal phenol concentration			
Fecal indole concentration			
Shannon index	56 (15 + 23 + 18)	60 (15 + 27 + 18)	[42,50,52]
Richness	92 (15 + 24 + 12 + 23 + 18)	96 (15 + 24 + 12 + 27 + 18)	[42,45,48,50,52]
Abundance of major bacterial phyla (Fusobacteria, Firmicutes, Actinobacteria, Bacteroidetes, and Proteobacteria)	77 (24 + 12 + 23 + 18)	81 (24 + 12 + 27 + 18)	[45,48,50,52]

Regarding intestinal microbiota, the meta-analyses focused on alpha diversity (expressed as richness and the Shannon index) and the abundance of the most important phyla in the canine gut microbiota (Fusobacteria, Firmicutes, Actinobacteria, Bacteroidetes, and Proteobacteria).

Alpha diversity consists of richness and evenness, from which various indices can be calculated, such as the Shannon index (H) or Faith’s phylogenetic diversity index [7]. Although these indices provide a more comprehensive analysis of alpha diversity, only the Shannon index could be examined across three studies [42,50,52], whereas richness (expressed as the number of detected operational taxonomic units [OTUs]) could be compared across five studies [42,45,48,50,52].

The abundance of bacterial phyla, expressed as the percentage of each phylum relative to the total bacterial population, was analyzed based on four studies [45,48,50,52]. While additional studies among the 13 identified articles assessed the microbiota composition, they used different measurement units, such as the logarithm of colony-forming units (CFU) [41] or changes relative to baseline [42].

For all the parameters specified above, the mean levels obtained from different studies were analyzed for both the control group and the postbiotic-supplemented group. Additionally, differences between both groups were examined to determine the effect of postbiotic use on these parameters.

The statistical analysis was conducted using Jamovi software (version 2.6.23). To analyze the mean levels of each parameter for both groups (control and supplemented), heterogeneity among studies was assessed using the Q test (statistical significance set at *p* < 0.05). Publication bias was also evaluated using the Fail-Safe N analysis, Egger’s test, and Begg and Mazumdar Rank Correlation Test (statistical significance set at *p* < 0.05).

Regarding the analysis of differences between groups (control vs. supplemented), both heterogeneity (Q test, *p* < 0.05) and publication bias (Fail-Safe N, Egger’s test, and Begg and Mazumdar Rank Correlation Test, *p* < 0.05) were assessed. Additionally, a random-effects model was applied to determine statistically significant differences (*p* < 0.05) between groups for each parameter.

## 3. Type of Postbiotics

### 3.1. Short-Chain Fatty Acids

Short-chain fatty acids (SCFAs) are organic acids primarily derived from the fermentation of polysaccharides by intestinal bacteria. Prebiotics (mainly FOS, MOS, and inulin) serve as fermentation substrates for these microorganisms, leading to SCFA production [53]. However, these postbiotics can also be synthesized from protein-based substrates, such as low-molecular-weight peptides [54].

SCFAs include acetic, propionic, and butyric acids, as well as their corresponding salts (acetate, propionate, and butyrate) [55]. Although their concentration varies throughout the intestine, the highest levels are found in the colon [56]. In feces, acetic acid accounts for 60% of SCFAs, while propionic and butyric acids each represent 20% [55]. These postbiotics are primarily produced by lactic acid bacteria, including *Lactobacillus acidophilus*, *Lactobacillus fermentum*, *Lactobacillus paracasei* ATCC 335, and *Lactobacillus brevis* [57].

Among the three acids, butyrate is the most significant due to its role as an energy source for enterocytes, supporting proper intestinal epithelium renewal and reinforcing intestinal barrier function [56] (Figure 6). Butyrate also possesses anti-inflammatory and immunomodulatory properties, primarily by inhibiting the transcription factor NF-κB1 [58].

**Figure 6 microorganisms-13-01572-f006:**
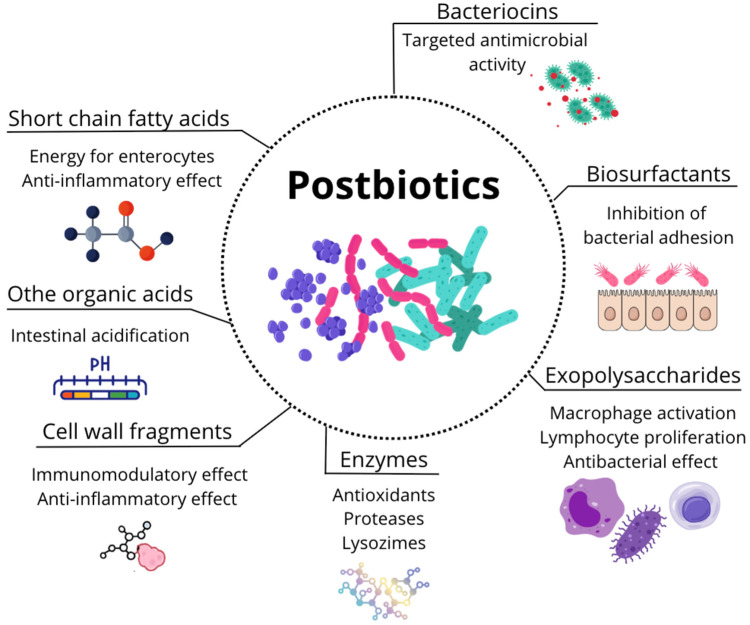
Overview of the main types of postbiotics and their mechanisms of action. Postbiotics include a diverse group of bacterial products—such as short-chain fatty acids, cell wall fragments, exopolysaccharides, bacteriocins, organic acids, biosurfactants, and enzymes—that contribute to host health [38,39]. These compounds exert various biological effects, including modulation of the immune system [59,60], antimicrobial activity [61,62], intestinal acidification [63], antioxidant and anti-inflammatory properties [64,65], and support for epithelial barrier integrity and nutrient absorption [56].

Both butyrate and, to a greater extent, propionate serve as substrates for hepatic gluconeogenesis [56] and exhibit antineoplastic activity, promoting apoptosis of malignant cells in the intestine [66]. These two molecules also play a role in erythropoiesis by enhancing duodenal iron absorption. This effect is mediated through the inhibition of hypoxia-inducible factor 2α (HIF2α), which regulates the production of ferroportin, duodenal cytochrome b, and divalent metal transporter 1 [67,68].

On the other hand, acetate has been shown to increase insulin sensitivity in humans [69] and improve resistance to enterohemorrhagic *Escherichia coli* O157:H7 infection while simultaneously reducing intestinal epithelial permeability to the toxins produced by this bacterium [70].

### 3.2. Other Organic Acids

In addition to short-chain fatty acids, other commonly bacterially derived organic acids include citric, tartaric, malic, and—most notably—lactic acid. These molecules exert antimicrobial effects by acidifying the surrounding environment, which inhibits bacterial growth by reducing intracellular pH and disrupting bacterial membrane integrity [71,72]. Under these acidic conditions, non-tolerant bacteria expend significant amounts of energy attempting to expel protons from their cytoplasm, ultimately leading to cell death [73].

The most commonly used bacterium for organic acid production is *Lactobacillus plantarum*, which can synthesize both D- and L-isomers of lactic acid. In in vitro cultures, the organic acids produced by *L. plantarum* have demonstrated inhibitory effects on pathogenic bacteria, such as *E. coli* and *Salmonella* spp. [64]. Therefore, the use of these postbiotics appears to promote gut health by acting as antimicrobial agents.

### 3.3. Cell Wall Fragments

The bacterial cell wall is composed of a wide variety of proteins, phospholipids, peptidoglycans, glycoproteins, and other molecular components [74]. Some of these components, when obtained as cell wall fragments, have been shown to exert beneficial effects on intestinal microbiota and overall mammalian health. This includes teichoic and lipoteichoic acids, which account for approximately 60% of the cell wall mass in Gram-positive bacteria, making them one of their predominant components [75]. These molecules exhibit immunomodulatory and anti-inflammatory effects in the host. By interacting with Toll-like receptor 2 (TLR2) on host immune cells, these compounds reduce interleukin-12 (IL-12) production while increasing the secretion of anti-inflammatory cytokines, such as interleukin-10 (IL-10). They also decrease levels of tumor necrosis factor-alpha (TNF-α), a pro-inflammatory cytokine induced by lipopolysaccharides [76,77].

Another cell wall-derived molecule of interest as a postbiotic is peptidoglycan. This molecule is recognized by NOD-like receptors (NLRs), specifically, NOD1, and NOD2, expressed on dendritic cells. Through this interaction, peptidoglycan promotes the maturation and proliferation of CD4+ T lymphocytes, which in turn downregulate pro-inflammatory cytokine production, exerting an anti-inflammatory effect [77].

This category of postbiotics also includes surface-associated bacterial peptides. Some of these peptides reinforce epithelial barrier function in enterocytes, while others reduce the intestinal absorption of heavy metals or exhibit anti-inflammatory and anti-adhesive properties against specific pathogenic bacteria [78].

### 3.4. Exopolysaccharides

Exopolysaccharides are large carbohydrate molecules composed of either repeating units of a single type of monosaccharide (homopolysaccharides) or multiple types of monosaccharides (heteropolysaccharides) [79]. Bacteria can synthesize these macromolecules, which accumulate on their surfaces and play key roles in cellular communication and adhesion, thereby contributing to biofilm formation [59,80]. Exopolysaccharides produced by lactic acid bacteria (*Lactococcus*, *Leuconostoc*, *Streptococcus*, *Pediococcus*, and *Bifidobacterium*) are widely used in the food and pharmaceutical industries due to their stabilizing and emulsifying properties [81,82]. However, these compounds have also been shown to provide health benefits when used as postbiotics.

These substances stimulate immune responses, as they are recognized by dendritic cells and macrophage receptors, promoting the proliferation of natural killer (NK) cells and T lymphocytes [83]. Some of these molecules enhance macrophage phagocytic activity and induce the secretion of nitric oxide (NO) in vitro [84]. Additionally, exopolysaccharides exhibit anti-inflammatory properties by reducing the production of pro-inflammatory cytokines (IL-1β, IL-6, and TNF-α) while increasing IL-10 secretion, an anti-inflammatory cytokine [85,86].

In vitro studies have demonstrated that bacterially derived polysaccharides exert antibacterial effects, as their presence has been shown to inhibit the growth of various pathogenic bacteria, including *Escherichia coli*, *Staphylococcus aureus*, *Bacillus subtilis*, and *Salmonella typhimurium*, among others [60,87,88]. Moreover, exopolysaccharides have also exhibited antitumor and antioxidant properties in cell culture models [89].

A key example of an exopolysaccharide is polysaccharide A (PSA), which is recognized by TLR2 receptors on CD4+ T lymphocytes. This recognition promotes the differentiation of these lymphocytes into FoxP3+ regulatory T cells, which produce IL-10. Additionally, PSA has been shown to inhibit the Th17 pro-inflammatory response and restore the Th1/Th2 immune balance (the balance between the cellular immune response and humoral immune response) in murine models [90].

### 3.5. Bacteriocins

Bacteriocins are bacterially derived peptides with antimicrobial activity against bacteria that are not of the same strain as the bacteriocin-producing bacterium [91,92]. All bacterial species are capable of producing at least one type of bacteriocin [92], and they protect themselves from the antimicrobial effects of their own bacteriocins by expressing specific immunity proteins encoded within the same operon as the bacteriocin they counteract [93].

The most commonly used postbiotic bacteriocins are those produced by lactic acid bacteria (LAB). These bacteriocins are composed of 30–60 amino acids [94], are thermostable, and typically do not undergo post-translational modifications [61]. One of the most common mechanisms of action of LAB-derived bacteriocins is pore formation in the cell membrane of target bacteria, exerting an inhibitory effect at nanomolar concentrations [62]. Additionally, many bacteriocins prevent biofilm formation through various mechanisms, including inhibiting pilus motility, reducing virulence factor expression, and interfering with quorum sensing—a bacterial communication system essential for biofilm development [95].

Through these mechanisms, bacteriocins have been shown to effectively inhibit the growth of several major pathogenic bacteria, including *Staphylococcus aureus*, *Proteus* spp., *Enterococcus* spp., *Pseudomonas aeruginosa*, *Escherichia coli*, *Salmonella* spp., *Clostridium difficile*, and *Klebsiella pneumoniae* [96,97]. Furthermore, these compounds have demonstrated efficacy against multidrug-resistant strains of *S.* aureus, *Salmonella* spp., and *E. coli*, suggesting their potential as alternatives to conventional antimicrobials for infection control [61,97].

In this regard, in vitro administration of LAB-derived bacteriocins to multidrug-resistant *Enterococcus faecium* cultures has demonstrated inhibitory effects on bacterial growth. Moreover, the combined use of these postbiotics with tetracycline resulted in a synergistic effect, enhancing bacterial growth inhibition [98]. Similarly, postbiotics derived from *Lactobacillus casei*, *Lactobacillus plantarum*, and *Lactobacillus salivarius*—commensal bacteria found in wild boars (*Sus scrofa*)—were shown to inhibit the in vitro growth of *Mycobacterium bovis*, the pathogen responsible for tuberculosis, whose primary reservoir is wild boars. These findings highlight the potential role of bacteriocins in infection control within their host organisms [63].

LAB-derived bacteriocins generally exhibit a narrower spectrum of activity than antibiotics, meaning that beneficial bacteria within the host’s gut microbiota remain unaffected by their antimicrobial action. This represents a key advantage over traditional antimicrobials, as their use does not lead to gut dysbiosis [99]. Although toxicity studies in mammals have not yet been conducted, in vitro studies have shown that high concentrations of bacteriocins do not exhibit cytotoxic effects on eukaryotic cell cultures, suggesting that bacteriocins, as postbiotics, can help eliminate pathogenic bacteria without harming host enterocytes or negatively impacting commensal gut bacteria [99].

### 3.6. Biosurfactants

Biosurfactants are surface-active macromolecules synthesized by bacteria, which are either released into the extracellular environment or remain attached to the bacterial surface. These compounds exhibit amphiphilic properties, containing both a hydrophobic and a hydrophilic region, allowing them to position themselves at the interface between fluids with different polarities, such as intestinal water and the lipid membranes of bacteria or enterocytes. Bacterially derived biosurfactants are commonly glycolipids, polysaccharides, phospholipids, glycopeptides, and glycoproteins, with varying structural complexity depending on their monomer composition and branching [100,101].

Due to their ability to interact with bacterial and cellular membranes, biosurfactants prevent pathogenic bacteria from colonizing the intestinal epithelium, hinder biofilm formation, and disrupt pre-existing biofilms, thereby promoting gut health [102,103].

Moreover, protein-based biosurfactants, derived from microbial synthesis, possess an amino acid balance close to that of an ideal protein, providing the host with high biological value proteins. This contributes to growth, tissue maintenance, and repair in the host organism [104].

### 3.7. Enzymes

Bacteria can produce a wide variety of enzymes, among which antioxidant enzymes stand out, including catalase, glutathione peroxidase (GPx), superoxide dismutase (SOD), and NADH oxidase [56]. In vitro studies have demonstrated that lactic acid bacteria, such as *Lactobacillus fermentum*, exhibit antioxidant effects by reducing hydroxyl radical formation through the synthesis of these enzymes [65] (Figure 7).

Antioxidant enzymes neutralize free radicals generated during the normal metabolism of both bacterial and eukaryotic cells, as well as those produced by certain pathogenic bacteria as virulence factors. In this way, these enzymes exert antioxidant effects, protecting lipid membranes, proteins, carbohydrates, and nucleic acids from oxidative damage caused by free radicals [56,65].

Since oxidative stress is pro-inflammatory, its reduction can lead to decreased inflammation. Therefore, beyond their antioxidant activity, these enzymes also exhibit anti-inflammatory effects in the host. For instance, the administration of postbiotics derived from *Lactobacillus plantarum* has been shown to increase serum GPx levels in ruminants [105]. Additionally, in murine models, the use of *Lactobacillus* strains producing SOD and catalase improved clinical signs associated with chronic intestinal disease and Crohn’s disease [106,107].

Other bacterially derived enzymes of interest include proteases and lysozymes. Proteases are degradative enzymes that hydrolyze the bonds between amino acids. *Bacillus* spp. is the main bacterial genus responsible for protease production, which facilitates the breakdown of protein chains into simpler peptides, thereby enhancing amino acid absorption by the host [108].

Lysozymes, in turn, are cytoplasmic enzymes that hydrolyze the bonds between N-acetylglucosamine (NAG) and N-acetylmuramic acid (NAM) molecules in peptidoglycan chains. For this reason, they are also referred to as 1,4-β-d-N-acetylmuramidases. Lysozymes differ from bacteriocins in that they are enzymes rather than peptides; however, they also exert antimicrobial effects by hydrolyzing the peptidoglycans that form the bacterial cell wall, thereby enhancing the host’s immune response and resistance to infections [109].

**Figure 7 microorganisms-13-01572-f007:**
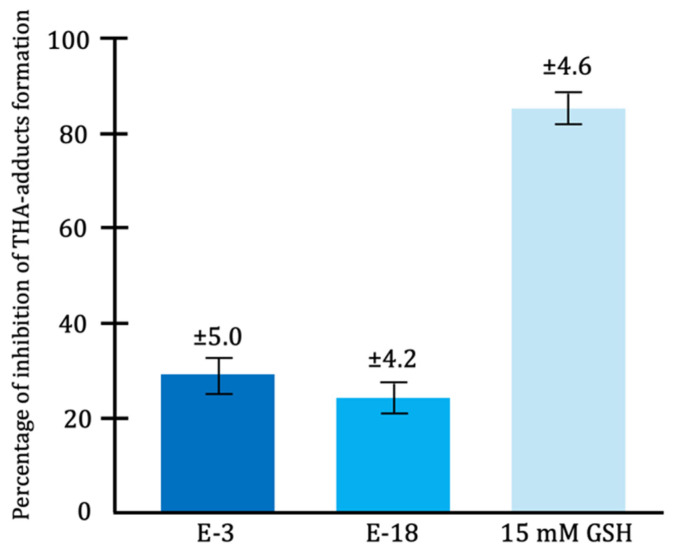
Antioxidant effect of *Lactobacillus fermentum* strains E-3 (dark blue) and E-18 (light blue), measured as percentage inhibition of terephthalic acid (THA) adduct formation. The antioxidant capacity of 15 mM reduced glutathione (GSH, very light blue) is also shown [65].

### 3.8. Other Metabolites

In addition to the postbiotics previously mentioned, bacteria produce a wide range of metabolites that also exhibit postbiotic activity. One such example is vitamins, which cannot be synthesized by the host but are essential for maintaining overall health, as they play key roles in numerous physiological functions [110].

Folic acid (vitamin B9) is one of the vitamins produced by bacteria used for postbiotic production and plays a crucial role in DNA synthesis, repair, and methylation while also being recognized for its antioxidant properties [56]. Another noteworthy vitamin is cyanocobalamin (vitamin B12), which is essential for proper hematopoiesis and neural health [111]. Studies have shown that supplementation with postbiotics derived from *Lactobacillus acidophilus* increases serum levels of vitamins B9 and B12, reducing the anemia prevalence in humans [112].

Similarly, vitamin K is essential for blood coagulation, bone health, and neuronal function, whereas riboflavin (vitamin B2) plays a key role in redox reactions as a hydrogen carrier. All these vitamins are synthesized by lactic acid bacteria and *Bifidobacterium* species, making them promising candidates for use as postbiotics to promote host health [103,110].

Bacteria used for postbiotic production also synthesize aromatic amino acids, which are bioactive compounds that influence organs beyond the intestine, including the brain, kidneys, and heart [113]. These molecules have shown beneficial effects in mice with chronic kidney disease [114].

Another key group of postbiotics includes polyphenols, such as urolithin A, equol, and 8-prenylnaringenin. Administration of urolithin A in mice for 10 weeks has been shown to reduce obesity and insulin resistance [115]. In humans, polyphenols have been associated with improved mitochondrial health; enhanced fatty acid oxidation; increased serum acylcarnitine levels [116]; and greater bone density, particularly in postmenopausal women [117].

Additionally, *Lactobacillus plantarum*, *Lactobacillus brevis*, and *Bifidobacterium* species synthesize several neurotransmitters, including serotonin, acetylcholine, and catecholamines (primarily dopamine and norepinephrine). These postbiotic molecules are involved in numerous neurological processes, such as mood regulation, motor control, memory, and learning [103,118].

Another molecule of particular interest is tryptophan, an amino acid and a precursor of serotonin. Tryptophan is degraded via the kynurenine pathway, which, when dysregulated, has been associated with inflammatory, neurodegenerative, depressive, and tumor-related processes in humans [67]. Furthermore, tryptophan is essential for protein synthesis and cell growth, influencing erythropoiesis. A 10% increase in tryptophan levels has been reported to raise hemoglobin and ferritin levels by 1.59% and 6.45%, respectively, thereby enhancing red blood cell production [119].

## 4. Mechanisms of Action and Health Effects in Dogs

### 4.1. Immunomodulatory Action

In in vitro studies and murine models, numerous postbiotic compounds have been shown to regulate immune system cells. For example, peptidoglycan promotes the maturation and proliferation of CD4+ T lymphocytes [77]. Exopolysaccharides enhance the functions of antigen-presenting cells, improve macrophage phagocytic activity, and promote the proliferation of T and NK lymphocytes [84]. Additionally, polysaccharide A (PSA) is recognized by CD4+ T lymphocytes, helping to restore the balance between cell-mediated and humoral immune responses [90].

A significant proportion of geriatric dogs suffer from chronic diseases, which can lead to chronic oxidative stress and accelerate cellular aging, including that of immune cells. This process negatively impacts immune responses by reducing the CD4+:CD8+ T-lymphocyte ratio—resulting in decreasing activation of CD4+ helper T cells and an increase in the CD8+ cytotoxic T-cells population. Consequently, immunosenescence, or aging of the immune system, disrupts the balance between Th1 and Th2 responses, making the host more susceptible to infections [47,120].

In a study [47], a postbiotic supplement containing short-chain fructooligosaccharides (scFOS) derived from yeast was administered to healthy geriatric dogs to evaluate its effects on their immune system. The supplemented dogs showed lower serum IgA concentrations compared to the control group, suggesting that the postbiotic promoted greater local IgA immune responses in mucosal tissues. Furthermore, supplemented dogs exhibited a higher CD4+:CD8+ T-lymphocyte ratio, indicating that scFOS supplementation may help counteract T-cell immunosenescence in dogs.

In another study [48], the administration of fermentation products from *Saccharomyces cerevisiae* to healthy adult dogs increased the population of monocytes, antigen-presenting B lymphocytes (MHC class II+ B cells), helper T cells, and IFN-γ-producing cytotoxic T cells compared to the control group. Dogs that received the postbiotic also exhibited higher serum IgE levels. Additionally, immune cells extracted from these dogs produced lower levels of TNF-α when stimulated with Toll-like receptor (TLR) agonists targeting TLR2, TLR3, TLR4, and TLR7/8.

In dogs exposed to stress factors (such as daily physical exercise and transportation), supplementation with postbiotics derived from *Saccharomyces cerevisiae* helped modulate immune responses, preventing significant alterations during recovery from exercise and stressful periods. The use of these products prevented an increase in blood leukocyte counts, particularly lymphocytes and eosinophils [49].

Atopic dermatitis is a chronic inflammatory and pruritic skin disease that affects 10–15% of dogs. This condition is associated with environmental allergens and tends to manifest more severely in body areas with higher allergen exposure. Affected dogs often exhibit elevated IgE levels, and additional factors such as skin barrier defects, oxidative stress, secondary infections, immune system disorders, and gut microbiota alterations also contribute to the disease pathogenesis [117].

In a study [50], researchers evaluated a supplement containing probiotics (*Lactobacillus rhamnosus*, *Bifidobacterium bifidum*, *Bifidobacterium infantis*, *Bifidobacterium animalis*, *Lactobacillus acidophilus*, and *Lactobacillus casei*), prebiotics (mannan-oligosaccharides and fructooligosaccharides), and a postbiotic obtained from *Saccharomyces cerevisiae* fermentation. The supplement was administered for 10 weeks to dogs with atopic dermatitis, and clinical improvements were assessed using the Pruritus Visual Analog Scale (PVAS10) and the Owner-Administered Skin Allergy Severity Index (OA-SASI), based on the Canine Atopic Dermatitis Extent and Severity Index (CADESI-4). Compared to the control group, supplemented dogs exhibited greater and earlier improvements in PVAS10 and OA-SASI scores, with PVAS10 values reaching normal ranges within four weeks in many cases.

In another study [51], researchers administered a 10-week supplementation of fermentation products from *Saccharomyces cerevisiae* to dogs with atopic dermatitis. By the end of the study, compared to the control group, supplemented dogs exhibited reduced transepidermal water loss in the ear region, higher sebum concentration, lower blood levels of activated T lymphocytes, and higher serum concentrations of the antioxidant enzyme superoxide dismutase (SOD). These findings led to clinical improvements in dogs with atopic dermatitis, suggesting that postbiotic supplementation may serve as a valuable adjunctive therapy for managing this condition.

### 4.2. Anti-Inflammatory Action

Postbiotics have demonstrated anti-inflammatory effects through various mechanisms. For example, butyrate inhibits the NF-κB1 transcription factor [58], while cell wall fragments and polysaccharide A (PSA) are recognized by TLR2 receptors, reducing the production of pro-inflammatory cytokines (IL-12 and TNFα) and increasing anti-inflammatory cytokines (IL-10) [76,77,90]. Similarly, exopolysaccharides promote IL-10 production while decreasing IL-1β, IL-6, and TNFα levels [85].

In a study where postbiotics derived from *Saccharomyces cerevisiae* were administered to dogs under stress [49], the supplemented group showed higher IL-10 levels (an anti-inflammatory cytokine) and maintained stable levels of IL-8 and monocyte chemoattractant protein-1 (both pro-inflammatory cytokines). In contrast, the control group experienced an increase in these pro-inflammatory cytokines following the stressful event. Similarly, another study [46] reported reductions of 7.85%, 23.97%, and 29.54% in serum levels of immunoglobulin G, TNFα, and interleukin-1, respectively, in adult dogs supplemented with postbiotics (Figure 8).

### 4.3. Modulatory Effects on the Gut Microbiota

Through different mechanisms, postbiotics can modulate the composition and diversity of the gut microbiota, which in turn impacts host health. In in vitro studies, the use of a postbiotic derived from *Limosilactobacillus reuteri* NBF 1 led to an increase in *Lactobacillus* spp. and *Bifidobacterium* spp.—bacteria beneficial to the host due to their ability to ferment prebiotics and produce short-chain fatty acids. This postbiotic also enhanced the populations of *Bacteroides* spp., *Prevotella* spp., *Porphyromonas* spp., *Staphylococcus* spp., and the Enterobacteriaceae family (including genera such as *Escherichia* spp., *Salmonella* spp., *Shigella* spp., *Citrobacter* spp., *Enterobacter* spp., *Erwinia* spp., *Klebsiella* spp., and *Proteus* spp.) [40].

Another study [41] administered the same postbiotic derived from *Limosilactobacillus reuteri* NBF 1 to sled dogs during competition to compare microbiota composition changes before and after the race in both the control and supplemented groups. After exercise, dogs receiving the postbiotic showed a lower increase in *E. coli* and *Streptococcus* spp.—bacteria considered enteropathogenic and indicators of dysbiosis. Conversely, these dogs exhibited a higher increase in bacteria from the genera *Faecalibacterium*, *Turicibacter*, *Blautia*, and *Fusobacterium*, as well as *Clostridium hiranonis*. These bacteria contribute to short-chain fatty acid production, providing health benefits to the host.

Similarly, administering fermentation products from *Saccharomyces cerevisiae* to physically stressed dogs increased *Turicibacter* spp. and *Lactobacillus* spp. populations compared to the control group, along with a reduction in *Clostridium* spp. [52]. Other studies have also shown that *S. cerevisiae* fermentation products positively affect the gut microbiota composition. In stressed dogs, supplementation with this type of postbiotic increased *Clostridium hiranonis* populations while reducing *Fusobacterium* spp. and *Blautia* spp. [49]. Another study found that *S. cerevisiae* fermentation products increased *Bifidobacterium* spp. abundance while reducing *Fusobacterium* spp. [48].

One study [50] investigated the use of this postbiotic in dogs with atopic dermatitis and found an increase in beneficial bacteria from the Firmicutes phylum, alongside a reduction in pathogenic bacteria from the Proteobacteria phylum. There was also an increase in Lachnospiraceae, a bacterial family involved in short-chain fatty acid production and intestinal barrier integrity, which is typically reduced in dogs with atopic dermatitis. Additionally, these dogs exhibited a rise in *Weissella* spp. (mainly *W. cibaria* and *W. confusa*), which appears to support the recovery of patients with skin and allergic diseases.

Another study [42] evaluated postbiotics derived from *Limosilactobacillus fermentum* and *Lactobacillus delbrueckii* in adult dogs. After five weeks, supplemented dogs showed greater alpha diversity, with increased counts of bacteria from the Actinobacteriota phylum and the genera *Faecalibaculum* spp., *Bifidobacterium* spp., and *Butyricicoccaceae* spp. However, the control group exhibited a higher increase in *Peptoclostridium* spp., *Sarcina* spp., and *Faecalitalea* spp. In a similar study, postbiotic supplementation boosted the abundance of beneficial bacteria like *Lactobacillus* spp. and *Bacillus* spp. while reducing harmful bacteria such as *Fusobacterium* spp. and *Anaerobiospirillum* spp. [46].

### 4.4. Antimicrobial Action

Postbiotics are effective in treating bacterial infections, offering a promising alternative to antibiotics in response to the increasing development of antimicrobial resistance [98]. The most important postbiotics with antimicrobial activity include bacteriocins, organic acids, and biosurfactants. These molecules, particularly bacteriocins, exert antimicrobial effects through various mechanisms, such as cell wall lysis, interference with quorum sensing, inhibition of biofilm formation, reduction of bacterial adhesion to the intestinal epithelium, and lowering intestinal pH [62,71,72,73,95,102,103].

In adult dogs, supplementation with postbiotics derived from *Bifidobacterium animalis* subsp. *lactis* CECT 8145 fermentation led to a decrease in intestinal pH to 5.8, compared to 6.1 in the control group. This reduction is beneficial, as it inhibits bacteria that cannot tolerate low-pH environments [43]. Another study [44] compared fecal properties among three groups of dogs: those receiving live *Bifidobacterium animalis* subsp. *lactis* CECT 8145 (probiotic), those supplemented with inactivated *Bifidobacterium animalis* subsp. *lactis* CECT 8145 (postbiotic), and an unsupplemented control group. Similar to the previous study, the feces of dogs receiving the postbiotic exhibited a lower pH and a higher concentration of propionate, which was associated with a healthier gut microbiota composition.

### 4.5. Antioxidant Action

The concept of postbiotics also includes bacterial enzymes, many of which have antioxidant properties, such as catalase, glutathione peroxidase (GPx), superoxide dismutase (SOD), and NADH oxidase [56,65]. In a study involving adult dogs, supplementation with fermentation products from *Saccharomyces cerevisiae* over 10 weeks increased serum concentrations of SOD and catalase [51]. Another study in adult dogs showed that administering a postbiotic supplement derived from *Limosilactobacillus fermentum* and *Lactobacillus delbrueckii* for five weeks resulted in a greater increase in serum superoxide dismutase levels compared to the control group [42].

### 4.6. Other Effects of Postbiotics

Bacteria produce a wide range of bioactive molecules and metabolites with postbiotic activity, many of which play roles in various metabolic processes. For example, in mice, postbiotic supplementation has been shown to reduce obesity and insulin resistance [117].

In dogs, supplementation with postbiotics has been associated with a 51.16% reduction in serum triglyceride levels, a 29.92% decrease in cholesterol levels, and an 11.43% decline in uric acid levels [46]. Furthermore, in dogs exposed to stressful events, supplementation with *Limosilactobacillus fermentum* and *Lactobacillus delbrueckii* resulted in smaller fluctuations in serum levels of corticosteroid isoenzyme alkaline phosphatase and alanine aminotransferase [42]. Additionally, dogs receiving postbiotics derived from *Bifidobacterium animalis* subsp. *lactis* CECT 8145 exhibited lower plasma concentrations of pancreatic polypeptide [43].

## 5. Meta-Analysis Results

Conducting a meta-analysis in a new research area is essential to know less evaluated aspects and to provide a context for future research. It allows us to combine findings from the existing studies, highlighting areas that require further investigation, aspects that need to be evaluated in detail, and different methodologies to follow. The main limitation is the small number of studies, which could produce biases; this is important to take into account. However, a meta-analysis of an early-stage topic, as is the use of postbiotics in dogs, is an essential step to determine further research.

### 5.1. Fecal Parameters

Table 3 presents the results of the mean value, heterogeneity (Q-test), and publication bias (Fail-Safe N, Begg and Mazumdar Rank Correlation, and Egger’s Regression) for all fecal parameters in both the control and postbiotic-supplemented groups. Additionally, Table 4 displays the estimated effects of postbiotic administration when comparing both groups, along with heterogeneity and publication bias results. Forest plots illustrating the differences between the control and postbiotic-supplemented groups for each parameter are provided in Appendix A).

As shown in these tables, despite the limited number of studies included in the meta-analysis, no statistically significant results (*p* > 0.05) were found regarding heterogeneity or publication bias for any fecal parameter, except for fecal butyrate concentration. This suggests that, except for this specific case, the results were homogeneous across studies, and no publication bias was detected, with no outliers or influential studies affecting result stability.

No significant differences (*p* > 0.05) were observed between the two groups for any fecal parameter. Regarding the fecal score, none of the analyzed studies [45,48,52] reported significant differences between groups. This may be because these experiments involved only clinically healthy dogs in both groups, all of whom had normal fecal scores at baseline. This stability in fecal scores is expected, as variations occur typically in pathological conditions, such as diarrhea or food allergies [121,122].

Although some postbiotic molecules, such as organic acids, lower the pH of their environment [71,72], the meta-analysis did not detect significant differences in fecal pH between groups. Among the analyzed studies [44,45,48,52], only [44] observed significant differences in fecal pH. This study administered postbiotics for the longest duration (90 days compared to 21 days in the other studies), suggesting that prolonged supplementation may be required to acidify the intestine and alter fecal pH.

Similarly, no statistically significant differences were found between groups regarding fecal concentrations of various compounds, including short-chain fatty acids (SCFAs: acetate, propionate, and butyrate), branched-chain fatty acids (BCFAs: isobutyrate, isovalerate, and valerate), indole, and phenol. When analyzing individual studies, significant changes related to postbiotic use were observed only in the reduction of fecal phenol and indole concentrations [48] and the increase in propionate levels [44].

SCFAs are produced by bacterial fermentation of polysaccharides in the gut [53], while BCFAs, indole, and phenol result from proteolytic fermentation by gut microbiota [45]. Postbiotic products contain these molecules, which, upon reaching the intestine, can be utilized by gut bacteria to complete metabolic pathways. For instance, SCFAs are rapidly absorbed by enterocytes, as they serve as their primary energy source. This utilization of fermentative products by both intestinal bacteria and epithelial cells may explain the lack of changes in fecal concentrations of these compounds following postbiotic administration [123].

### 5.2. Gut Microbiota

Table 5 presents the results of the mean value, heterogeneity (Q-test), and publication bias (Fail-Safe N, Begg and Mazumdar Rank Correlation, and Egger’s Regression) for all parameters related to gut microbiota in both the control and postbiotic-supplemented groups. Likewise, Table 6 shows the estimated effect of postbiotic administration on these parameters by comparing both groups, as well as the heterogeneity and publication bias results. Forest plots illustrating the differences between the control and postbiotic-supplemented groups for each parameter are provided in Appendix A).

Similar to the fecal parameters, despite the low number of studies included in the meta-analysis, no statistically significant results (*p* > 0.05) were found regarding heterogeneity and publication bias for any of these parameters, except for the abundance of the Fusobacteria and Firmicutes phyla. This suggests stability in the results.

Regarding alpha diversity (richness and Shannon index), the meta-analysis does not reveal significant differences between the control group and the postbiotic-supplemented group. However, when analyzing each study individually, greater statistical significance (*p* < 0.10) was observed in experiments where the postbiotic was administered for a longer period (5 weeks [42] and 10 weeks [50]), compared to the three weeks of supplementation in the other studies [48,52]. This suggests that a longer supplementation period with postbiotics is necessary to observe changes in alpha diversity. Additionally, in the study [50], where dogs with atopic dermatitis were used, postbiotic supplementation appeared to produce significant changes in the microbiota compared to healthy animals with a normal microbial population.

No statistically significant differences were detected in the relative abundance of any of the major phyla in the canine gut microbiota (Fusobacteria, Firmicutes, Actinobacteria, Bacteroidetes, and Proteobacteria). Two of the four studies included in this meta-analysis [45,52] did not detect significant changes in phylum abundance following supplementation. In contrast, [50] observed a significant increase in the Firmicutes phylum and a decrease in the Proteobacteria phylum. Meanwhile, [48] detected a significant increase in the Actinobacteria phylum and a reduction in the abundance of the Fusobacteria phylum.

Across these four studies [45,48,50,52], significant changes were observed in several bacterial genera, particularly, an increase in genera associated with SCFA production and a decrease in the abundance of pathogenic bacteria such as *Klebsiella pneumoniae* or *Streptococcus pasteurianus*. However, due to the heterogeneity in the genera studied across studies, a meta-analysis at the genus taxonomic level could not be performed.

The lack of significant results in the meta-analysis indicates that there is still no clear answer regarding the use of postbiotics in dogs, signaling the need for future research on this key topic. Our findings also provide insight into which parameters would be worth evaluating, highlighting the microbial composition of the gut and suggesting that, to find clear answers, future research should primarily focus on experimental studies in diseased individuals.

## 6. Production and Presentation Forms of Postbiotics

The production of postbiotics from bacterial or fungal populations is not a recent development. In the food industry, particularly in dairy products, fermentative processes using lactic acid bacteria are very common [110]. However, in this type of fermentation, neither the quantity nor the type of postbiotics produced is controlled, making their health effects on consumers less evident. For this reason, industrial postbiotic production typically involves the use of biofermenters. These are tanks with several liters of capacity in which optimal conditions are established for the growth of the target bacterial strain: aerobic or anaerobic conditions, a temperature suitable for the species, and a nutrient-rich substrate necessary for its development (proteins, polysaccharides, etc.) [111].

The most used species in postbiotic production belong to the genera *Lactobacillus*, *Streptococcus*, *Bifidobacterium*, *Eubacterium*, *Saccharomyces*, and *Faecalibacterium* [111]. During the fermentation process, these microorganisms produce a wide variety of bioactive postbiotic compounds, such as short-chain fatty acids, organic acids, exopolysaccharides, and proteins [56]. Furthermore, through biofermentation processes, it is possible to control both the type and concentration of postbiotics produced by adjusting incubation time, the type of fermentation substrate, and the bacterial strain used. As a result, products obtained through this method have demonstrated anti-inflammatory, antioxidant, and immunomodulatory properties in the host consuming these postbiotics [124,125].

Once the fermentation process is complete, postbiotics can be extracted in different ways. One type of product is cell-free supernatant (CFS), a liquid containing the metabolites produced during microbial fermentation and soluble nutrients that were not consumed, but without cellular debris. Therefore, CFS contains fatty acids, organic acids, and protein molecules that exert antioxidant, anti-inflammatory, and antitumor effects on the consumer [56,126].

Another method of postbiotic production involves inactivating the microorganisms used in biofermentation. Various bacterial cell wall lysis techniques can be employed for this purpose, such as enzymatic lysis, heat lysis, sonication, or high-pressure lysis [127]. These treatments allow for the extraction of not only soluble metabolites but also intracellular metabolites and bacterial structural components, resulting in what is known as bacterial lysates [111,128].

Once obtained, postbiotics can be marketed in liquid form or, more commonly, in solid form, typically as lyophilized (freeze-dried) products [129]. The viability of these products can be affected by various environmental factors, such as humidity, storage temperature, or oxidation. Therefore, it is essential to apply specific techniques to ensure their stability. For solid products, capsules with a semi-permeable coating are used to isolate postbiotic substances from external conditions and ensure their release at a specific point in the gastrointestinal tract. These coatings are usually made from plant-based biopolymers (e.g., cellulose, pectin, gum arabic, alginate, inulin, etc.), animal-based biopolymers (e.g., chitin, hyaluronic acid, etc.), or microbial-origin polymers (e.g., xanthan, dextran, gellan gum, etc.). Proteins from animal sources (e.g., caseins, albumins, whey proteins, etc.) or plant sources (e.g., soy or pea protein) may also be used. For liquid products, UV-protected glass containers are commonly used to prevent solar radiation penetration. Emulsions can also be used to help stabilize the product [130].

## 7. Future Perspectives

The meta-analyses conducted in this study did not yield a statistically significant expected effect from the use of postbiotics on fecal parameters and the gut microbiota composition. Although the statistical tests performed to assess heterogeneity and publication bias did not show significant values (*p* > 0.05), it is important to note that the different meta-analyses carried out for each parameter were based on only 2–4 studies. It would be advisable to have more studies available to validate these results. In most of these studies, postbiotics were administered to healthy dogs for approximately three weeks. However, when these products were administered to sick dogs [50] or over a longer period (about 9–12 weeks) [44,50], the effect of postbiotics showed greater statistical significance in the analyzed parameters. Additionally, some authors suggest that higher doses of postbiotics may be necessary to better evaluate their effects in these cases [45,52].

From the systematic literature review, not enough studies (only one or two examples) were found to conduct a meta-analysis on other parameters, such as specific bacterial genera, inflammatory markers, or cellular and humoral immune responses. Furthermore, there is a lack of studies evaluating the evolution of blood test results (hematology, biochemical panel) in dogs supplemented with postbiotics. Therefore, it would be beneficial to conduct further research on postbiotic supplementation in dogs, particularly in those suffering from intestinal or immune-mediated diseases, to measure not only fecal and microbial parameters (analyzing variations in the abundance of major bacterial families and genera) but also immune, inflammatory, and serological markers. Moreover, these studies should be conducted over periods longer than three weeks and with higher doses of postbiotics.

Despite these limitations, available results from both in vitro experiments and studies in humans and other animal species demonstrate the potential benefits of these compounds. Postbiotics have been shown to improve gastrointestinal health through various mechanisms of action, such as their anti-inflammatory and immunomodulatory effects [58,85,90], antimicrobial activity [71,73,98,103], and ability to regulate the gut microbiota composition [48,49,50,52]. However, due to the close relationship between the gut and the rest of the body, as well as the absorption of some of these bacterial metabolites [56,111], postbiotics may also enhance the health of other organs, including the kidneys [20], heart [24], nervous system [19], skin [17], and joints [18,40,41].

As a result, postbiotics represent a promising tool for treating common canine diseases, such as gastrointestinal disorders (acute diarrhea or chronic inflammatory bowel disease [IBD]), inflammatory and immune-mediated conditions (atopic dermatitis, osteodegenerative disease, food allergies, bacterial infections, parasitic diseases, etc.), kidney diseases, metabolic and endocrine disorders (primarily diabetes), and behavioral disorders. Although preliminary results on their effects in healthy dogs are limited, future research should focus on patients with the aforementioned conditions.

Postbiotics also offer a promising solution in the fight against antimicrobial resistance. This is mainly due to the activity of bacteriocins produced by certain bacterial strains, which have been shown to effectively inhibit the growth of major pathogenic bacteria, such as *Escherichia coli* and *Salmonella* spp. Furthermore, the use of these compounds in combination with antibiotics has demonstrated a synergistic effect in the treatment of infections caused by multidrug-resistant bacteria [93].

A common practice in small animal clinics is the administration of probiotics alongside antibiotics to reduce the harmful effects of antibiotics on gut microbiota, preventing intestinal dysbiosis [36]. However, most bacterial species used as probiotics have been shown to be susceptible to common antibiotics [56]. Therefore, their co-administration with antibiotics remains controversial, as antibiotics inhibit probiotics, preventing them from exerting their regulatory effect on the gut microbiota composition [131]. In contrast, the activity of postbiotics is not diminished by antibiotics. This is because postbiotics are already synthesized bacterial compounds, meaning their concentration and efficacy are not reduced by antimicrobial molecules, allowing them to be effective in reducing dysbiosis caused by antibiotic administration [42]. Additionally, since probiotics are composed of live bacteria, they colonize the host’s intestine and can exchange genetic material with resident bacteria through horizontal gene transfer. This poses a risk for the transfer of antimicrobial resistance genes, as studies have already demonstrated the presence and transfer of such genes from probiotic bacterial strains to commensal bacteria [132]. In contrast, because postbiotics are bacterial components or non-viable bacteria, they do not exchange genetic material with the host’s gut microbiota [56].

The exclusive administration of postbiotics offers numerous advantages for improving canine well-being and treating various diseases [42,47,50]. However, their use in combination with conventional treatments for certain conditions could be particularly interesting, as a synergistic effect is expected [133]. Combining postbiotics with prebiotics and probiotics could also enhance their beneficial effects, as seen in synbiotics [134]. For example, both postbiotics and probiotics have been shown to reduce the severity of skin lesions in dogs with atopic dermatitis, suggesting that their combined administration could have a synergistic effect in these cases [50,51].

In addition to their numerous benefits for overall health and specific organs and systems, postbiotics are considered safe for administration [56,126]. None of the reviewed studies on postbiotic use in humans or animals reported any adverse effects. Moreover, bacteria commonly used for postbiotic production have received Qualified Presumption of Safety (QPS) status, as they are generally regarded as posing no health risks [111]. In many cases, the genomes of these bacteria are sequenced to search for resistance or virulence genes [126]. Therefore, although further safety and toxicity studies are necessary, the use of these compounds appears to be safe [110].

For these reasons, further development in the field of postbiotics is highly valuable, both in terms of their production and the development of new strains capable of producing these metabolites, as well as in the formulation of various food products containing them. By expanding research on their effects on overall health and exploring their potential combination with pharmaceutical treatments or other bioactive components (prebiotics and probiotics), postbiotics may aid in the management and treatment of chronic diseases affecting a significant portion of the pet population, such as osteoarthritis, atopic dermatitis, diabetes, and IBD.

This systematic review and meta-analysis have several limitations that should be acknowledged. First, although the search strategy was designed to identify studies on postbiotic interventions, it relied primarily on the term postbiotic and did not explicitly include alternative descriptors such as “inactivated microorganisms” or “non-viable microbes.” This decision was made to avoid retrieving a large volume of irrelevant records, such as studies on vaccines or pathogen inactivation, but may have led to the exclusion of some relevant articles that used pre-2021 terminology. Second, the decision to combine studies involving different microbial species and canine populations was driven by the scarcity of available evidence. While most included studies shared common features—such as similar inactivation methods and comparable experimental designs—the heterogeneity in interventions and populations introduces potential sources of bias. This includes a risk of underestimating effects due to dilution across differing contexts (i.e., bias toward the null). We have attempted to mitigate this through appropriate weighting based on sample size and by limiting the meta-analysis to parameters reported in multiple studies. Nevertheless, we recognize that this exploratory synthesis should be interpreted with caution and viewed as a preliminary step to inform future, more targeted analyses as the evidence base grows. Despite these limitations, such preliminary syntheses are crucial for identifying knowledge gaps and informing the design of future hypothesis-driven studies in this emerging field.

## 8. Conclusions

This systematic literature review explores the different types of postbiotics, their mechanisms of action, and the effects of their administration on canine health. These microbial-derived substances help regulate the immune system, inflammatory response, and gut microbiota composition, which in turn influences overall health and aids in the management of various systemic diseases. The meta-analysis conducted did not yield statistically significant results regarding the use of postbiotics. This lack of significant results highlights the novelty of the research field and the need for further investigation, given the potential importance of postbiotics in canine health. Further research is recommended to explore the application of postbiotics in sick dogs, establishing administration guidelines for the treatment of various diseases, and to investigate their potential in combination with other bioactive compounds, such as prebiotics or probiotics.

## Figures and Tables

**Figure 5 microorganisms-13-01572-f005:**
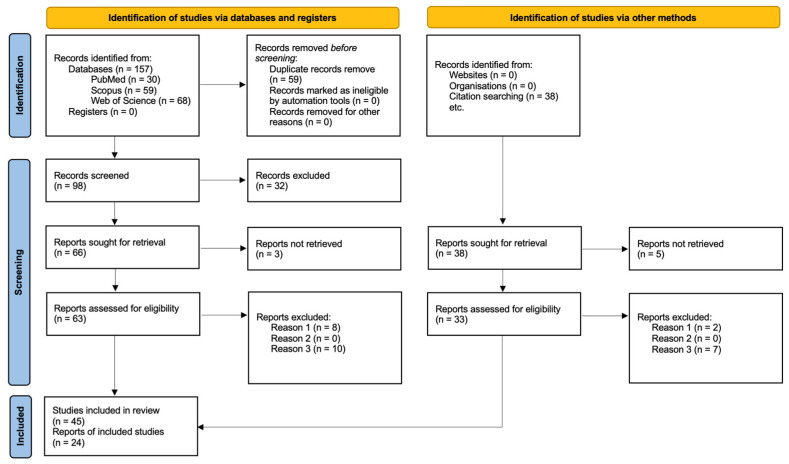
PRISMA flow diagram of the systematic literature review conducted in PubMed, Web of Science, and Scopus between 22 September 2024 and 10 October 2024.

**Figure 8 microorganisms-13-01572-f008:**
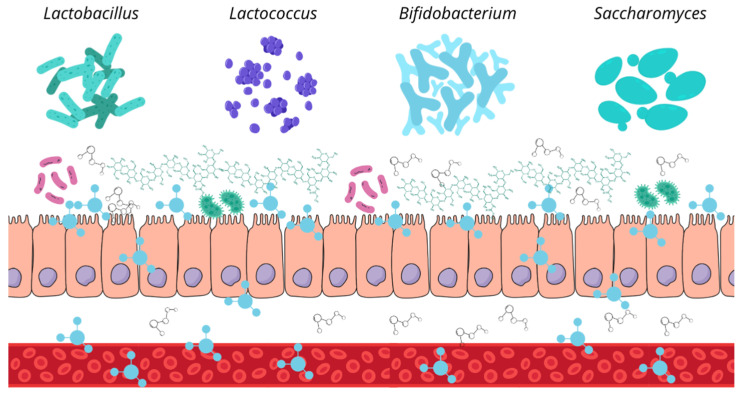
The biofermentation of various microorganisms allows producing a wide variety of postbiotics. Supplementing dogs with these molecules provides intestinal benefits but also systemic advantages, as many of these compounds are transported via the bloodstream.

**Table 3 microorganisms-13-01572-t003:** Mean value, heterogeneity, and publication bias results for each fecal parameter in both the control and postbiotic-supplemented groups.

Parameter	Group	Mean Value	Q-Test	*p*	Fail-Safe N	*p*	Begg and Mazumdar Rank Correlation	*p*	Egger’s Regression	*p*
Fecal score	Control	3.13	0.0004	0.9998	0.000	0.419	−1.000	0.333	−0.019	0.985
	Supplemented	3.17	0.0003	0.9999	0.000	0.418	−1.000	0.333	−0.015	0.988
pH	Control	6.29	0.0007	1.000	0.000	0.324	0.183	0.718	0.013	0.990
	Supplemented	6.32	0.0008	1.000	0.000	0.324	−0.183	0.718	0.001	0.999
Acetate ^1^	Control	311	1.0220	0.7959	4.000	0.013	0.183	0.718	−0.082	0.935
	Supplemented	313	0.9112	0.8227	4.000	0.012	−0.183	0.718	−0.073	0.942
Propionate ^1^	Control	160	0.2018	0.9773	0.000	0.128	−0.183	0.718	−0.125	0.901
	Supplemented	167	0.1839	0.9801	0.000	0.117	−0.183	0.718	−0.114	0.910
Butyrate ^1^	Control	61.05	3.1650	0.3669	25.000	<0.001	0.183	0.718	−0.204	0.839
	Supplemented	70.65	5.1391	0.1619	35.000	<0.001	0.183	0.718	0.378	0.705
Isobutyrate ^1^	Control	5.75	0.0082	0.9998	0.000	0.333	0.913	0.071	0.085	0.932
	Supplemented	5.80	0.0079	0.9998	0.000	0.333	0.548	0.279	0.076	0.940
Isovalerate ^1^	Control	8.67	0.0169	0.9994	0.000	0.258	0.913	0.071	0.122	0.903
	Supplemented	8.93	0.0189	0.9993	0.000	0.253	0.548	0.279	0.103	0.918
Valerate ^1^	Control	3.14	0.0593	0.9962	0.000	0.401	0.548	0.279	0.133	0.895
	Supplemented	4.51	0.1052	0.9912	0.000	0.362	0.548	0.279	0.159	0.874
Phenol ^1^	Control	1.58	0.0226	0.9991	0.000	0.448	0.913	0.071	0.094	0.925
	Supplemented	1.41	0.0182	0.9994	0.000	0.454	0.913	0.071	0.075	0.941
Indole ^1^	Control	1.49	0.0009	1.0000	0.000	0.455	0.913	0.071	0.030	0.976
	Supplemented	1.42	0.0006	1.000	0.000	0.458	0.913	0.071	0.024	0.981

^1^ The concentration of these molecules has been quantified in all studies as μmol/g of fecal dry matter.

**Table 4 microorganisms-13-01572-t004:** Estimated effect of postbiotic administration (difference from the control group), heterogeneity, and publication bias results for each fecal parameter.

Parameter	Estimated Effect	*p*	Q-Test	*p*	Fail-Safe N	*p*	Begg and Mazumdar Rank Correlation	*p*	Egger’s Regression	*p*
Fecal score	0.0946	0.624	1.064	0.587	0.000	0.268	1.000	0.333	0.954	0.340
pH	−0.1052	0.621	4.647	0.200	0.000	0.242	−1.000	0.083	−1.606	0.108
Acetate	0.1287	0.463	3.178	0.365	0.000	0.213	0.333	0.750	0.434	0.664
Propionate	0.2802	0.276	6.086	0.107	0.000	0.058	0.333	0.750	1.005	0.315
Butyrate	0.8601	0.361	32.035	<0.001	6.000	0.007	0.667	0.333	5.089	<0.001
Isobutyrate	0.0302	0.862	0.221	0.974	0.000	0.445	−0.667	0.333	−0.235	0.814
Isovalerate	0.1017	0.560	0.684	0.877	0.000	0.296	0.000	1.000	−0.276	0.782
Valerate	0.3887	0.124	5.863	0.118	3.000	0.015	0.667	0.333	1.357	0.175
Phenol	−0.1400	0.424	2.326	0.508	0.000	0.158	−1.000	0.083	−1.377	0.168
Indole	−0.1381	0.429	1.414	0.702	0.000	0.182	−0.667	0.333	−0.822	0.411

**Table 5 microorganisms-13-01572-t005:** Alpha diversity results (richness and Shannon index) and abundance of the main bacterial phyla (Fusobacteria, Firmicutes, Actinobacteria, Bacteroidetes, and Proteobacteria) in both the control and postbiotic-supplemented groups.

Parameter	Group	Mean Value	Q-Test	*p*	Fail-Safe N	*p*	Begg and Mazumdar Rank Correlation	*p*	Egger’s Regression	*p*
Richness ^1^	Control	142	0.1282	0.9980	0.000	0.110	−0.200	0.817	0.003	0.998
	Supplemented	139	0.0861	0.9991	0.000	0.107	0.000	1.000	0.058	0.954
Shannon index	Control	4.06	0.0030	0.9985	0.000	0.389	0.333	1.000	0.039	0.969
	Supplemented	4.11	0.0047	0.9977	0.000	0.382	0.333	1.000	0.062	0.951
Fusobacteria ^2^	Control	23.10	0.3484	0.9507	2.000	0.028	0.000	1.000	0.446	0.656
	Supplemented	20.52	0.1548	0.9845	1.000	0.042	0.333	0.750	0.309	0.758
Firmicutes ^2^	Control	61.31	1.3877	0.7084	29.000	<0.001	−1.000	0.083	−1.149	0.251
	Supplemented	63.21	1.0321	0.7935	34.000	<0.001	−0.667	0.333	−0.941	0.346
Actinobacteria ^2^	Control	1.59	0.0019	1.0000	0.000	0.451	−0.333	0.750	−0.016	0.987
	Supplemented	3.35	0.0302	0.9986	0.000	0.387	0.667	0.333	0.072	0.943
Bacteroidetes ^2^	Control	16.75	0.2647	0.9665	0.000	0.080	1.000	0.083	0.446	0.656
	Supplemented	17.14	0.2467	0.9697	0.000	0.072	0.333	0.750	0.358	0.720
Proteobacteria ^2^	Control	5.43	0.0092	0.9998	0.000	0.328	1.000	0.083	0.093	0.926
	Supplemented	5.50	0.0082	0.9998	0.000	0.322	0.333	0.750	0.074	0.941

^1^ Richness is expressed as the number of OTUs isolated in the fecal sample. ^2^ The abundance of different phyla is expressed as a percentage of the total microbiota.

**Table 6 microorganisms-13-01572-t006:** Estimated effect of postbiotic administration (difference from the control group), heterogeneity, and publication bias results for each parameter related to gut microbiota.

Parameter	Estimated Effect	*p*	Q-Test	*p*	Fail-Safe N	*p*	Begg and Mazumdar Rank Correlation	*p*	Egger’s Regression	*p*
Richness	−0.0900	0.539	2.631	0.621	0.000	0.313	0.400	0.483	0.909	0.363
Shannon index	0.0254	0.892	1.242	0.537	0.000	0.400	1.000	0.333	1.114	0.265
Fusobacteria	−0.1476	0.355	0.413	0.938	0.000	0.178	0.000	1.000	−0.062	0.951
Firmicutes	0.0571	0.720	0.274	0.965	0.000	0.347	0.333	0.750	0.285	0.775
Actinobacteria	0.1150	0.471	0.513	0.916	0.000	0.251	−0.333	0.750	−0.313	0.755
Bacteroidetes	0.0574	0.719	0.146	0.986	0.000	0.377	−0.333	0.750	−0.314	0.754
Proteobacteria	0.0360	0.821	0.653	0.884	0.000	0.431	−0.333	0.750	−0.359	0.720

## Data Availability

The protocol for this systematic literature review was registered on the Open Science Framework (OSF) and is available at https://doi.org/10.17605/OSF.IO/BMNAF (accessed on 30 May 2025). For further details related to the systematic review or the meta-analysis, please refer to the Appendix A or contact dbonel@ucm.es.

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
