# Peer review of "Effects of Postbiotic Administration on Canine Health: A Systematic Review and Meta-Analysis"

_microorganisms, 2025, doi:10.3390/microorganisms13071572_

Round 1

Reviewer 1 Report

Comments and Suggestions for Authors

The review systematically expounded the effects of postbiotic on canine, which is very meaningful and important to the present studies. There are a little problems as follows,

L 44, .....1012 ....? Lack of CFU here?

L104, Figure1, I wonder whether this was a result from two refereces? For in different references, the ration of different bacteria were totally different for the different conditions. I suggest using Table here to show the different ratio of bacteria. Also, generally ,we usually focus on phylum and genus instead of orders.

Figure 2 , As gut-liver, gut-kidney, gut-brain were widely studied on mice or humans, are there any related canine studies about these axis? If have, please show.

Figure 3, showed the abundance of bacteria orders in the gut microbiota of dogs of different healthy status, but I noticed that reference [15] just showed the fecal micriobiome without gut microbiota, please check.

Figure 4, please show the related references of the definition. Also, organic acids include SCFAs.

L327 & Table1, please check the format of the cited references.

Table 1, If have, please add the dosage of the postbiotics.

Why the order of references was not according to their order of appearance? For example, [104-116] were in front of [38,39, .....]

Figure 6 , part contents about postbiotic of the figure were repeated with figure 4. Please modify.

Figure 7 was the result of [41]?or [89]? The figure was confused here.

Part 3.1 &3.5 could be integrated.

Part 3.7 just showed the antioxidant enzymes, but other funtional enzymes, like lysozyme, protease etc. not mentioned here.

Part 2.2 showed meta-analysis, but 5 also showed, which was confused.

L788, p should be italic and consistent, or capital, or lowercase letter.

Author Response

Response to Reviewer 1 Comments

We would first like to sincerely thank the reviewer for their valuable comments and thoughtful revision of our manuscript. We fully agree with all the points raised and have revised the paper accordingly. Detailed responses to each comment are provided below, and the corresponding changes have been clearly highlighted in the resubmitted files (using track changes or indicated as appropriate). We remain open and willing to implement any further modifications the reviewer may deem necessary in order to improve the quality and clarity of the manuscript.

Point-by-point response to Comments and Suggestions for Authors

Comments 1: L 44, .....1012 ....? Lack of CFU here?

Response 1: Thank you very much for your comment. You are absolutely right in noting the importance of specifying the unit of measurement. In this case, we have opted to use "bacterial cells" rather than colony-forming units (CFU) because the figure of approximately 10¹² bacteria per gram refers to the total estimated number of bacterial cells, not solely those that are culturable (line 44).

Comments 2: L104, Figure1, I wonder whether this was a result from two refereces? For in different references, the ration of different bacteria were totally different for the different conditions. I suggest using Table here to show the different ratio of bacteria. Also, generally ,we usually focus on phylum and genus instead of orders.

Response 2: We sincerely thank the reviewer for this helpful observation. The inclusion of two references in the description of Figure 1 was an oversight — the figure was in fact adapted solely from reference [14]. In that study, the authors compare the relative abundance of bacteria at the order level; although we recognize that it is more common to present microbiota composition at the phylum or genus level, the original publication does not provide data at those taxonomic ranks. Therefore, we included the figure as it appears in the original source to visually represent the main trends reported (line 108).

Comments 3: Figure 2 , As gut-liver, gut-kidney, gut-brain were widely studied on mice or humans, are there any related canine studies about these axis? If have, please show.

Response 3: Thank you once again for your valuable comment. The various gut-organ axes are indeed discussed later in the manuscript, specifically in the context of studies conducted in dogs (lines164-207). To improve clarity, we have now added the corresponding references to the caption of Figure 2 to explicitly indicate that these axes have been demonstrated in the canine species (lines 135-139).

Comments 4: Figure 3, showed the abundance of bacteria orders in the gut microbiota of dogs of different healthy status, but I noticed that reference [15] just showed the fecal micriobiome without gut microbiota, please check.

Response 4: Thank you very much for your insightful comment. Upon revisiting the references, we confirmed that although references [17] and [19] refer to the gut microbiota, the analyses were in fact conducted on fecal samples. Therefore, we have updated the figure caption to specify that the data represent the fecal microbiota (lines 160-163).

Comments 5: Figure 4, please show the related references of the definition. Also, organic acids include SCFAs.

Response 5: Thank you very much for your careful observation. We have added the appropriate references and revised the figure. While SCFAs are indeed classified as organic acids, we have chosen to separate them in the figure to emphasize their particular importance in relation to gut microbiota and intestinal health. For this reason, SCFAs are listed individually, followed by a separate category labeled "other organic acids" to encompass the remaining compounds (lines 249-254).

Comments 6: L327 & Table1, please check the format of the cited references.

Response 6: Thank you very much for your comment. As suggested, we have adapted the reference style to conform to the journal’s formatting requirements (Table 1, Table 2, lines 378-383).

Comments 7: Table 1, If have, please add the dosage of the postbiotics.

Response 7: We appreciate the reviewer’s insightful comment. In Table 1, we have added a column indicating the daily dosage of postbiotics, as well as another column specifying the duration of supplementation. However, this information was not available in all studies; in many cases, the authors reported the amount of dietary supplement administered, but did not specify the exact mass of inactivated microbial cells.

Comments 8: Why the order of references was not according to their order of appearance? For example, [104-116] were in front of [38,39, .....]

Response 8: Thank you very much for your observation. We fully agree, and for this reason, we have revised and reordered the references to ensure they now follow their correct order of appearance in the text.

Comments 9: Figure 6 , part contents about postbiotic of the figure were repeated with figure 4. Please modify.

Response 9: Thank you very much for your comment. We have modified Figure 6 to eliminate the overlap with Figure 4. The revised Figure 6 now focuses on illustrating the main functions of the different types of postbiotics (line 480).

Comments 10: Figure 7 was the result of [41]?or [89]? The figure was confused here.

Response 10: Thank you very much for pointing this out. Since citation [46] (formerly [41]) was already mentioned in the paragraph, we have removed it from the figure caption to clarify that the figure is derived from the results reported in reference [104] (formerly [89]) (lines 661-663).

Comments 11: Part 3.1 &3.5 could be integrated.

Response 11: Thank you very much for your suggestion. As also reflected in Figure 4, we have renamed Section 3.5 as Section 3.2 but have kept SCFAs and other organic acids as separate groups due to the distinct and significant role SCFAs play, despite being part of the organic acid group. However, we have clarified in the text that SCFAs are indeed a subset of organic acids (lines 501-512).

Comments 12: Part 3.7 just showed the antioxidant enzymes, but other funtional enzymes, like lysozyme, protease etc. not mentioned here.

Response 12: Thank you very much for your insightful comment. Following your suggestion, we have added new paragraphs describing the health benefits of other functional enzymes, including proteases and lysozymes (lines 648-659).

Comments 13: Part 2.2 showed meta-analysis, but 5 also showed, which was confused.

Response 13: Thank you very much for your observation. Section 2.2 describes the methodology used to conduct the meta-analysis, whereas Section 5 presents the results obtained. To clarify this distinction, we have retitled Section 5 as Meta-analysis Results (line 871).

Comments 14: L788, p should be italic and consistent, or capital, or lowercase letter.

Response 14: Thank you very much for your observation. We have corrected the formatting throughout the manuscript to ensure that p is consistently presented in italics and lowercase.

Once again, we are truly grateful for the reviewer’s time, effort, and constructive feedback, which have greatly contributed to enhancing our work.

Reviewer 2 Report

Comments and Suggestions for Authors

This manuscript presents a well-conducted systematic review and meta-analysis concerning the effects of postbiotic administration on canine health. The research is of considerable importance, contributing to our understanding of postbiotic effects and applications in dogs, as well as broader issues related to canine microbiome and gut health. The manuscript is generally well-written, demonstrating a strong command of English, a logical structure, and appropriate use of current references. While the lack of significant findings is acknowledged, the experimental design and data analysis appear sound and relevant. To further strengthen the manuscript, I suggest the authors consider:

1. The introduction section is overly lengthy and contains tangential information. For instance, lines 35-91 delve into a detailed exploration of general microbiology and the human microbiome that is too basic and lacks direct relevance to the canine-specific focus of this manuscript. I recommend significantly condensing or removing this material to improve the introduction’s clarity and focus.

2. To enhance reproducibility and accessibility, please provide direct links to each database utilized in this study. This would be particularly helpful in the materials and methods section. For example, add the URL for each database mentioned on lines 264-267.

3. Please ensure consistent and correct formatting of p-values throughout the manuscript. Refer to the journal’s guidelines for specific formatting requirements.

4. The current citation method detracts from the readability of the manuscript. Please incorporate references directly after each specific cited claim or piece of information, rather than listing a long string of references at the end of a paragraph. This applies to the following sections:L35-41, L72-78, L109-115, L164-193, L877-889, L933-941, L971-977.

5. Additionally, the following sections lack sufficient references and require the addition of citations to support the claims made: L56-62, L79-88, L95-101, L867-872, L942-954, L980-994.

With attention to these minor issues, I recommend this manuscript for publication.

Author Response

Response to Reviewer 2 Comments

We would first like to sincerely thank the reviewer for their valuable comments and thoughtful revision of our manuscript. We fully agree with all the points raised and have revised the paper accordingly. Detailed responses to each comment are provided below, and the corresponding changes have been clearly highlighted in the resubmitted files (using track changes or indicated as appropriate). We remain open and willing to implement any further modifications the reviewer may deem necessary in order to improve the quality and clarity of the manuscript.

Point-by-point response to Comments and Suggestions for Authors

Comments 1: The introduction section is overly lengthy and contains tangential information. For instance, lines 35-91 delve into a detailed exploration of general microbiology and the human microbiome that is too basic and lacks direct relevance to the canine-specific focus of this manuscript. I recommend significantly condensing or removing this material to improve the introduction’s clarity and focus.

Response 1: Thank you very much for your valuable comment. We understand your concern regarding the length and broader focus of the introduction. However, we respectfully consider that this section is crucial for providing essential background from a One Health perspective. While the core of our study is canine-specific, the interconnectedness between human, animal, and environmental microbiota is fundamental for contextualizing the broader implications of microbiome research. We believe that introducing the topic in this way is beneficial, as it provides a clear narrative thread that facilitates coherence and enhances the overall readability of the manuscript. Nonetheless, if the reviewer deems it essential to shorten this section, we would be more than happy to make the necessary adjustments.

Comments 2: To enhance reproducibility and accessibility, please provide direct links to each database utilized in this study. This would be particularly helpful in the materials and methods section. For example, add the URL for each database mentioned on lines 264-267.

Response 2: Thank you very much for your suggestion. We agree with your comment and have added the URLs for each database accordingly (lines 293-294).

Comments 3: Please ensure consistent and correct formatting of p-values throughout the manuscript. Refer to the journal’s guidelines for specific formatting requirements.

Response 3: Thank you once again for your valuable comment. We have reviewed the manuscript to ensure that all p-values are consistently formatted in italics and lowercase, as specified in the journal’s guidelines.

Comments 4: The current citation method detracts from the readability of the manuscript. Please incorporate references directly after each specific cited claim or piece of information, rather than listing a long string of references at the end of a paragraph. This applies to the following sections:L35-41, L72-78, L109-115, L164-193, L877-889, L933-941, L971-977.

Response 4: Thank you very much for your comment. In accordance with your suggestion, we have added in-text citations to the mentioned sections so that readers can more clearly understand the source of the information presented (lines 35-41, 72-78, 109-115, 1000-1008, 1050-1054, 1088-1103).

Comments 5: Additionally, the following sections lack sufficient references and require the addition of citations to support the claims made: L56-62, L79-88, L95-101, L867-872, L942-954, L980-994.

Response 5: Thank you very much for your comment. We fully agree and have added the necessary citations to the indicated sections to strengthen and support the claims made (lines 56-62, 79-88, 95-101, 988-922, 1066-1074, 1100-1111).

Once again, we are truly grateful for the reviewer’s time, effort, and constructive feedback, which have greatly contributed to enhancing our work.

Reviewer 3 Report

Comments and Suggestions for Authors

The manuscript is excessively long and much of it is a literature review (albeit well-done) of all non-antibiotic "biotics", not just prebiotics. As the authors day, when it comes to post-biotics in dogs there isn't much to say. I think this endeavor is  premature if the authors stick to the premise of the title - there are too few data to conclude anything.
The review is well written and may deserve publication in an alternative journal or a simple "review". If its just postbiotics it could be much shorter!
I really have no comments for the authors other than I do not think there is enough new material to warrant publication. I think the editorial staff need to advise the authors as they think fit.

Author Response

Response to Reviewer 3 Comments

We would first like to sincerely thank the reviewer for their valuable comments and thoughtful revision of our manuscript. We remain open and willing to implement any further modifications the reviewer may deem necessary in order to improve the quality and clarity of the manuscript.

Point-by-point response to Comments and Suggestions for Authors

Comments 1: The manuscript is excessively long and much of it is a literature review (albeit well-done) of all non-antibiotic "biotics", not just prebiotics. As the authors day, when it comes to post-biotics in dogs there isn't much to say. I think this endeavor is  premature if the authors stick to the premise of the title - there are too few data to conclude anything.

The review is well written and may deserve publication in an alternative journal or a simple "review". If its just postbiotics it could be much shorter!

I really have no comments for the authors other than I do not think there is enough new material to warrant publication. I think the editorial staff need to advise the authors as they think fit.

Response 1: Thank you very much for your comment. We understand your concern regarding the length and scope of the manuscript. However, our intention was to provide a detailed and structured synthesis of all available information related to postbiotics and their potential applications in dogs, within the broader and increasingly relevant context of gut microbiota modulation in veterinary medicine.

Although data specific to dogs are currently limited, this scarcity highlights the importance of an early and comprehensive review. By compiling the existing studies and performing a meta-analysis, we aim to identify current knowledge gaps and suggest key areas for future research, such as the evaluation of hematological parameters (including red and white blood cell counts) and microbiota profiling across different taxonomic levels.

Furthermore, we believe that establishing a solid conceptual framework now is critical for guiding future research, avoiding redundant efforts, and promoting the rational design of studies assessing the efficacy and safety of postbiotics. In this way, our review contributes not only to summarizing the current state of the field but also to fostering its advancement.

Once again, we are truly grateful for the reviewer’s time, effort, and constructive feedback, which have greatly contributed to enhancing our work.

Round 2

Reviewer 3 Report

Comments and Suggestions for Authors

My original comments still stand. It is an editorial decision in my opinion as to whether my opinion matters or not for this journal. I appreciate your well written review.